# The MpsAB Bicarbonate Transporter Is Superior to Carbonic Anhydrase in Biofilm-Forming Bacteria with Limited $CO_2$ Diffusion

Sook-Ha Fan,[a] Miki Matsuo,[a] Li Huang,[a,b] Paula M. Tribelli,[a,c] Friedrich Götz[a]

[a]Microbial Genetics, Interfaculty Institute of Microbiology and Infection Medicine Tübingen (IMIT), University of Tübingen, Germany
[b]Key Laboratory of Animal Disease and Human Health of Sichuan Province, Institute of Preventive Veterinary Medicine, Sichuan Agricultural University, Wenjiang, People's Republic of China
[c]Departamento de Química Biológica, FCEyN-UBA, Buenos Aires, Argentina

**ABSTRACT**   $CO_2$ and bicarbonate are required for carboxylation reactions, which are essential in most bacteria. To provide the cells with sufficient $CO_2$, there exist two dissolved inorganic carbon supply (DICS) systems: the membrane potential-generating system (MpsAB) and the carbonic anhydrase (CA). Recently, it has been shown that MpsAB is a bicarbonate transporter that is present not only in photo- and autotrophic bacteria, but also in a diverse range of nonautotrophic microorganisms. Since the two systems rarely coexist in a species but are interchangeable, we investigated what advantages the one system might have over the other. Using the genus *Staphylococcus* as a model, we deleted the CA gene *can* in *Staphylococcus carnosus* and *mpsABC* genes in *Staphylococcus aureus*. Deletion of the respective gene in one or the other species led to growth inhibition that could only be reversed by $CO_2$ supplementation. While the *S. carnosus* Δ*can* mutant could be fully complemented with *mpsABC*, the *S. aureus* Δ*mpsABC* mutant was only partially complemented by *can*, suggesting that MpsAB outperforms CA. Interestingly, we provide evidence that mucus biofilm formation such as that involving polysaccharide intercellular adhesin (PIA) impedes the diffusion of $CO_2$ into cells, making MpsAB more advantageous in biofilm-producing strains or species. Coexpression of MpsAB and CA does not confer any growth benefits, even under stress conditions. In conclusion, the distribution of MpsAB or CA in bacteria does not appear to be random as expression of bicarbonate transporters provides an advantage where diffusion of $CO_2$ is impeded.

**IMPORTANCE**   $CO_2$ and bicarbonate are required for carboxylation reactions in central metabolism and biosynthesis of small molecules in all bacteria. This is achieved by two different systems for dissolved inorganic carbon supply (DICS): these are the membrane potential-generating system (MpsAB) and the carbonic anhydrase (CA), but both rarely coexist in a given species. Here, we compared both systems and demonstrate that the distribution of MpsAB and/or CA within the phylum *Firmicutes* is apparently not random. The bicarbonate transporter MpsAB has an advantage in species where $CO_2$ diffusion is hampered—for instance, in mucus- and biofilm-forming bacteria. However, coexpression of MpsAB and CA does not confer any growth benefits, even under stress conditions. Given the clinical relevance of *Staphylococcus* in the medical environment, such findings contribute to the understanding of bacterial metabolism and thus are crucial for exploration of potential targets for antimicrobials. The knowledge gained here as exemplified by staphylococcal species could be extended to other pathogenic bacteria.

**KEYWORDS** *Staphylococcus aureus*, *Staphylococcus carnosus*, *Firmicutes*, MpsAB, carbonic anhydrase, biofilm

Address correspondence to Friedrich Götz, friedrich.goetz@uni-tuebingen.de.

CO₂ and bicarbonate ($HCO_3^-$) are simple carbon molecules and yet are so important in the life of prokaryotes and eukaryotes. The classical $CO_2$ fixation is carried out by many autotrophic organisms/organelles, such as chloroplasts and the cyanobacterial carboxysomes. They possess ribulose-1,5-bisphosphate carboxylase/oxygenase (RuBisCO), which catalyzes the fixation of inorganic $CO_2$ to ribulose-1,5-bisphosphate (RuBP) to form two molecules of 3-phosphoglycerate (1). Since $CO_2$ is gaseous and thus can diffuse in and out of the cell, it must be trapped in the cell to be available in sufficient quantity for carboxylation reactions. Many autotrophic and also nonautotrophic microorganisms use carbonic anhydrase (CA) enzymes as a dissolved inorganic carbon supply (DICS) system (2, 3). CAs are ubiquitous enzymes that can be found in the mitochondria, cytoplasm, periplasm, membrane, or cell-wall-associated carboxysome and also chloroplast in plants (3, 4). Most of these enzymes have a Zn-binding domain that catalyzes the reversible interconversion of $CO_2$ and bicarbonate ($CO_2 + H_2O \leftrightarrow HCO_3^- + H$). CA is important for inorganic carbon-fixing enzymes that utilize either $CO_2$ or $HCO_3^-$ by interconverting these species to ensure a sufficient concentration in the cytoplasm. To date, CAs from eight evolutionary distinct families have been reported ($\alpha$, $\beta$, $\gamma$, $\delta$, $\zeta$, $\eta$, $\theta$, and $\iota$) (5–8). Many prokaryotes possess putative CA genes of two or more families, whereas some even possess multiple genes from the same family, suggesting that this enzyme plays an important role in prokaryotic physiology and fitness (3).

Apart from the CA-catalyzed cytoplasmic interconversion of $CO_2$ to $HCO_3^-$, there is another mechanism, which is based on a membrane-localized $HCO_3^-$ transporter. Such a transporter transports $HCO_3^-$ from the external environment over the membrane into the cytoplasm, where the imported $HCO_3^-$ is consumed by the carboxylation reactions. The continuous consumption of $HCO_3^-$ in the cytoplasm could induce a suction power to keep the transporter running. In the exterior milieu, the transporter is continuously removing $HCO_3^-$ from the $CO_2/HCO_3^-$ equilibrium, resulting in a permanent replenishment of $HCO_3^-$. Unlike $CO_2$, $HCO_3^-$ is charged, which prevents an immediate back-diffusion over the membrane. In this regard, most cellular systems depend on dedicated transporters to facilitate the movement of membrane-impermeant $HCO_3^-$ across the cell membrane (9). However, such transporters have until recently been described only in autotrophic bacteria, such as in cyanobacteria, where the mechanisms for DIC transport are well characterized. DIC refers to the total amount of $CO_2$, $HCO_3^-$, and $CO_3^{2-}$ in water. Given the absence of a classical bicarbonate uptake system, questions arise as to how other autotrophs and some nonautotrophic bacteria lacking these transporters and/or CAs manage to concentrate intracellular $HCO_3^-$.

A recent resurgence of interest in bicarbonate transporters appears to be driven largely by the discovery by the group of Kathleen M. Scott of a novel two-component transporter that facilitates dissolved inorganic carbon (DIC) uptake (10). They were the first to describe that these transporters possess DIC uptake activity in *Hydrogenovibrio crunogenus*, a chemolithoautotroph from gammaproteobacteria and confirmed to transport DIC (11). Subsequently, these homologs which were named DABs, were also reported to accumulate $HCO_3^-$ in a sulfur-oxidizing gammaproteobacterial chemolithoautotroph, *Halothiobacillus neapolitanus* (12). Surprisingly, DAB homologs were already described earlier in the nonautotrophic bacterium *Staphylococcus aureus*, where they were referred to as MpsAB (membrane-potential generating system) because they contribute to the membrane potential generation (13). Further studies showed that MpsAB represents an $HCO_3^-$-concentrating system, possibly acting as a sodium bicarbonate cotransporter (14), the first example of such transporters in the phylum *Firmicutes*. Subsequently, we used an *S. aureus* HG001 *mpsABC* deletion mutant for most of the experiments, but follow-up studies showed that *mpsC* is not a functional part of the *mpsAB* operon. It was further found that all examined representatives of the species *S. aureus* possess only the MpsAB transporter, but no CA.

In our previous study, we reported that MpsAB and CA represent a $CO_2$/bicarbonate-concentrating system, each of which can functionally replace the other (14). Against this background, we aimed to understand the link between both the systems

and to investigate if one system is superior than the other by using staphylococcal strains. In *S. aureus* HG001, MpsAB is essential for growth under atmospheric air while no CA-encoding gene is present. On the other hand, a gene annotated as encoding a putative CA from *Staphylococcus carnosus* TM300 has yet to be investigated for its physiological function or enzymatic activity.

Here, we report that the distribution of MpsAB and/or CA within the phylum *Firmicutes* is apparently not random. The bicarbonate transporter MpsAB has an advantage in species where $CO_2$ diffusion is hampered—for instance, in mucus and biofilm-forming bacteria.

## RESULTS

**The distribution of MpsAB and/or CA in selected *Firmicutes* species.** First, we wanted to get an overview of the distribution of the two completely different DICS systems, MpsAB and/or CA, within the phylum *Firmicutes*. A selection of families within this phylum carrying MpsAB, CA(s), or both based on the presence and/or absence of Pfam motif is shown in Table 1. Due to the huge numbers of strains available in the database, we limited our search to only the finished genomes in the IMG/M database (15), and only one representative strain per species is shown in Table S1 in the supplemental material. Almost all of the species representatives have at least one system—either MpsAB or CA (Table 1; Table S1). Nevertheless, quite a number of species, particularly in the *Bacillaceae* family, have both systems. A similar observation was seen only in *S. sciuri* (*Staphylococcaceae*) and *Sulfobacillus acidophilus* (*Clostridiaceae*). It is interesting to note that many species have two copies of CA genes, of either the prokaryotic (Pfam00484) or eukaryotic (Pfam00194) type of CA. We use the terms $\beta$-CA for Pfam00484 and $\alpha$-CA for Pfam00194 in order to more adequately describe the evolutionary history of these enzymes. It seems that $CO_2$-concentrating systems play an important role in bacteria, and the presence of two systems is probably advantageous in certain habitats.

To gain an insight into their genome organization, we then looked at the comparative synteny maps of various *Firmicutes* species (see Fig. S1 in the supplemental material). Examples of staphylococcal strains harboring only MpsAB are *S. aureus*, *S. epidermidis*, and *S. haemolyticus* (14). In these genomes, a hypothetical protein, MpsC, is located immediately downstream of MpsAB (Fig. S1A). MpsC was initially thought to be part of the Mps operon, but further analysis revealed that it is not a functional part of it (14). Next, we investigated the CA localization in genomes containing both MpsAB and $\beta$-CA (Fig. S1B). In *S. sciuri*, *Bacillus anthracis*, and *Bacillus subtilis*, a putative CA is located immediately downstream of MpsAB homologs. The other CAs of *S. sciuri* and *B. subtilis* located elsewhere in the genome shared a much higher identity with the *S. carnosus* CA (Fig. 1B). In the two *Bacillus* strains, the second CA gene is located between *luxS* and *cydA* genes, whereas *B. subtilis* has a third CA gene located elsewhere. In the third part, genomes harboring only CAs can be further divided into those with $\beta$- or $\alpha$-CAs (Fig. S1C). Among the $\beta$-CAs, the *Staphylococcaceae* family members share a similar location for CA. It is always located downstream of a gene (labeled as mp in Fig. S1C) that encodes a membrane protein of unknown function. No common synteny was observed in the other group of CAs, suggesting that CA localization in the genomes is conserved only in closely related species or genera.

**The CA from *S. carnosus* and its functional complementation in *E. coli* Δ*can*.** As we identified the staphylococcal CA genes based on Pfam domain, we first need to establish whether the identified genes encode active CA enzymes. We used *S. carnosus* TM300 and *S. pseudintermedius* ED99 as our model strains to investigate CA in more detail. The annotation of the genome of *S. carnosus* TM300 (hereafter *S. carnosus*) and *S. pseudintermedius* ED99 (hereafter *S. pseudintermedius*) revealed that a single putative CA-encoding gene, SCA_1457 and SPSE_0869, respectively, is present in each (16, 17). To verify that no other potential CAs are present, the deduced *can* gene product of the respective staphylococcal strains were subjected to a BLAST search (BLASTp) against their own genome sequences. No significant homology to any other proteins was

**TABLE 1** The presence of MpsAB and/or CA in selected *Firmicutes*

| Family | Species | Presence of MpsAB or CA (PFam)[a] | | |
|--------|---------|------|--------|--------|
| | | MpsAB | $\beta$-CA (Pfam00484) | $\alpha$-CA (Pfam00194) |
| *Bacillaceae* | *Anoxybacillus* sp. strain B2M1 | + | − | − |
| | *Bacillus anthracis* Ames | + | + | − |
| | *Bacillus anthracis* BF1 | − | + | − |
| | *Bacillus subtilis* ATCC 13952 | − | + + | − |
| | *Bacillus subtilis spizizenii* W23 | + | + + | − |
| | *Bacillus thuringiensis* 97-27 | + | + | − |
| | *Geobacillus kaustophilus* HTA426 | + | − | − |
| | *Lysinibacillus macroides* DSM 54 | − | + | − |
| | *Oceanobacillus iheyensis* CHQ24 | − | + | − |
| *Listeriaceae* | *Listeria monocytogenes* NCTC7973 | − | − | + |
| *Staphylococcaceae* | *Macrococcus caseolyticus* IMD0819 | − | + | − |
| | *Staphylococcus aureus aureus* USA300_FPR3757 | + | − | − |
| | *Staphylococcus carnosus* TM300 | − | + | − |
| | *Staphylococcus epidermidis* RP62A | + | − | − |
| | *Staphylococcus pseudintermedius* ED99 | − | + | − |
| | *Staphylococcus sciuri* SNUSD-18 | (+) | + | − |
| *Enterococcaceae* | *Enterococcus faecalis* KB1 | − | − | + |
| | *Enterococcus faecium* ERS2704476 | − | − | + |
| *Lactobacillaceae* | *Lacticaseibacillus casei* ATCC 334 | − | − | + |
| | *Lactiplantibacillus plantarum* 16 | − | − | + |
| | *Lactobacillus acidophilus* FSI4 | − | − | − |
| | *Lactobacillus delbrueckii* JCM 17838 | − | − | + |
| *Streptococcaceae* | *Lactococcus lactis* subsp. *lactis* 14B4 | − | − | − |
| | *Lactococcus piscium* CMTALT02 | − | − | + + |
| | *Streptococcus pneumoniae* NCTC 7466 | − | + | − |
| | *Streptococcus pyogenes* M28PF1 | − | + | − |
| | *Streptococcus salivarius* 57.I | − | + | + |
| | *Streptococcus suis* DN13 | − | + | − |
| | *Streptococcus thermophilus* LMG 18311 | − | + | + |
| *Clostridiaceae* | *Clostridium botulinum* 111 | − | + | − |
| | *Clostridium perfringens* CP15 | − | + | − |
| | *Sulfobacillus acidophilus* TPY | + | + | − |

[a]The presence of the proteins was inferred based on the following Pfam domain search from finished bacterial genomes in the Integrated Microbial Genomes & Microbiomes (IGM/G) database: MpsAB (Pfam00361 and Pfam10070, respectively), prokaryotic type-carbonic anhydrase (Pfam00484), and eukaryotic-type CA (Pfam00194). Other Pfam domains, such as Pfam08936 for carboxysome shell carbonic anhydrase (CsoSCA), Pfam18484 for cadmium CA repeat, and Pfam10563 for a putative CA-like domain, were also searched within the above *Firmicutes*, but these domains were not found. The terms $\beta$-CA for Pfam00484 and $\alpha$-CA for Pfam00194 are used in the table in order to more adequately describe the evolutionary history of these enzymes. The symbols + and − indicate the presence or absence, respectively, of the protein domains. The symbol +/− indicates the presence or absence of the protein domains, with + indicates one and + + indicate two protein domains. The symbol (+) indicates that in *S. sciuri* SNUSD-18, MpsA and MpsB appear to be truncated.

found, implying that only a single $\beta$-CA is present in each strain. The 192-amino-acid CA from *S. carnosus* shares 70% identity with the CA from *S. pseudintermedius* (193 amino acids) and has a comparable identity of 28% to *Streptococcus pneumoniae* TIGR4 and, to a lesser extent, *E. coli* MG1655 at 26% (Fig. 1B). For *S. carnosus* and *S. pseudintermedius* genomes, we did not find any other homology (BLASTp) with $\alpha$-CAs from *Enterococcus faecium*, *Helicobacter pylori*, *Neisseria gonorrhoeae*, and *Vibrio cholerae* and human CA1 and CA2, as well as $\gamma$-CAs from *Enterococcus faecium*, *Escherichia coli*, *Halobacterium salinarum*, and *Methanosarcina thermophila*.

According to the NCBI Conserved Domain database, the *S. carnosus* CA belongs to the $\beta$-class and D clade (cd03379). Likewise, *Streptococcus pneumoniae* and *E. coli* are also members of the $\beta$-CAs (18, 19). Multiple-sequence alignment revealed that *S. carnosus*, along with the bacteria in Fig. 1A, shares a highly conserved motif. Such distinctive motif is consistent with those found in $\beta$-CAs, namely, the HXXC motif (where H is

## A

```
                                                           38 40 42
carnosus            ---MTLLESILAYNKDF----VGN--KEFENYTTSKKPDKKAVLFTCMDTRLQDLGTKAL    51
pseudintermedius    ---MTLLEHILEYNEEF----VAN--KAYEAYSTSKTPSKKAVLLTCMDTRLQDLSTKAL    51
sciuri              ---MPLLNDILEYNAAF----IEN--KEYENLITTKTPNAKAVLLTCMDTRLTELSTRAL    51
subtilis_CA1        ---MSLLNDILEFNKTF----TEQ--REYEKYQTSKFPDKKMAILSCMDTRLVELLPHAM    51
subtilis_CA2        MNQMVSLTSILEHNQRF----VSE--KKYEPYKTTKFPSKKLVIVTCMDTRLTELLPQAM    54
Str.pneumoniae      ---MSYFEQFMQANQAY----VAL--HG--QLNLPLKPKTRVAIVTCMDSRLH--VAQAL    47
E.coli              ---MKDIDTLISNNALWSKMLVEEDPGFFEKLA--QAQKPRFLWIGCSDSRVP--AERLT    53
                         *  :  ::  *   :               . :   . * *:*:       :

                                                      96   99
carnosus            GFNNGDLKVVKNAGAIITHPYGSTIKSLLVGIYALGAEEIIIMAHKDCGMGCLDVSTVKD   111
pseudintermedius    GFNNGDLKVVKNAGATISHPFGSTMRSLLVGIYALGAEEIIIMGHKDCGMGNINVEEVMT   111
sciuri              GFKNGDIKVVKNAGATISHPYGSTMRSLLVAIYALGAEEIIIMGHKDCGMGNLNVDSVID   111
subtilis_CA1        NLRNGDVKIIKSAGALVTHPFGSIMRSILVAVYELNADEVCVIGHHDCGMSKISSKSMLE   111
subtilis_CA2        GLKNGDAKIVKNAGAIVSHPFGSVMRSILVAIYELQAEEVCIVGHHECGMSGLNASSILE   114
Str.pneumoniae      GLALGDAHILRNAGGRVTE---DMIRSLVISQQQMGTREIVVLHHDCGAQTFENEPFQE   104
E.coli              GLEPGELFVHRNVANLVIHTDLNCLSVVQYAVDVLEVEHIIICGHYGCGGVQAAVEN---   110
                     .:   *:   :  :...    :   .  :   .  . .:  *    **       .

carnosus            AMKERGVTEETFKIIEHSGVDV-DSFLQGFKDAEE--------------NVRRNIDMVY--   155
pseudintermedius    TMQQRGVDEQVIDILNYSGIDV-SNFLKGFDDVYD--------------NVKHNIGMIY--   155
sciuri              TMKSRGITDDTLNTIEHSGINI-HQFLRGFDDVTE--------------NVQTNIQKVY--   155
subtilis_CA1        KIKARGIPEERIETIKYSGVDF-DQWFKSFDSVEA--------------SVKDSVDVIK--   155
subtilis_CA2        KAKERGVEDSCLNLLTSAGLDL-KTWLTGFHSVEE--------------SVSHSVNMIK--   158
Str.pneumoniae      YLKEE-----------LGVDVSDQDFLPFQDIEE--------------SVREDMQLLI--   137
E.coli              ---------PELGLINNWLLHIRDIWFKHSSLLGEMPQERRLDTLCELNVMEQVYNLGHS   161
                             :..     :          . *     .: :

carnosus            ----NHPLFDKSVPIHGLVIDPHTGELDLIQDGYELA----AQNK--------------   192
pseudintermedius    ----EHPLFDNKVPVHGLVIDPHTGELDLVHNGYERA----ESYQN-------------   193
sciuri              ----NHPLFDQSVPIHGLVIDPHNGDLEVIQNGYEFT----K-----------------   189
subtilis_CA1        ----HHPLFPENVPVHGLVIDPKTGKLDLIVNGYNN---------------------   187
subtilis_CA2        ----NHPLLPKKVPVHGLVIHPETGKLDVVINGYETE----LINNHS-----------   197
Str.pneumoniae      ----ESPLIPDDVIISGAIYNVDTGSMTVVEL---------------------------   165
E.coli              TIMQSAWKRGQKVTIHGWAYGIHDGLLRDLDVTATNRETLEQRYRHGISNLKLKHANHK   220
                          ..* : *       .  * :   :
```

## B

| Strains | Identity (%) | E-value |
|---|---|---|
| *S. carnosus* | 100 | - |
| *S. pseudintermedius* | 70 | 6e-103 |
| *S. sciuri* | 67 | 8e-100 |
| *B. subtilis CA1* | 47 | 1e-71 |
| *B. subtilis CA2* | 49 | 6e-74 |
| *Str. Pneumoniae* | 28 | 4e-23 |
| *E. coli* | 26 | 6e-07 |

**FIG 1** Multiple protein sequence alignment for CA. (A) The alignment was carried out for the CAs from *S. carnosus* TM300 (carnosus), *S. pseudintermedius* ED99 (pseudintermedius), *S. sciuri* FDAARGOS_285 encoded by locus tag Ga0225916_842 with Pfam00484 (sciuri_CA), *Bacillus subtilis* subtilis 168 encoded by locus tag BSU30690 with Pfam00484 (subtilis_CA1), *B. subtilis* 168 encoded by locus tag BSU34670 with Pfam00484 (subtilis_CA2), *Str. pneumoniae* TIGR4 encoded by locus tag SP0024 (pneumoniae), and *E. coli* MG1655 encoded by locus tag b0126 (*E. coli*). All sequences were obtained from the IMG/M database. The deduced sequences were aligned using Clustal Omega (56). The numbers correspond to the *S. carnosus* sequence. Highlighted in red are residues 38 (cysteine), 96 (histidine) and 99 (cysteine), which are ligands to zinc. Residues 40 and 42 highlighted in yellow are an Asp/Arg dyad that is important for the proton transfer step of catalysis. (B) The percentages of identity of the CA protein from *S. carnosus* to those of the other sequences were compared using BLASTp (51).

histidine, C is cysteine, and X is any residue), which binds to the active site metal ion, and the CXDXR motif (where C is cysteine, D is aspartic acid, R is arginine, and X is any residue), which completes the active site (20). A metal ion binding site that consists of one His and two Cys residues was found at residues 38, 96, and 99, a characteristic

**TABLE 2** Growth of the deletion mutant and its complemented strains

| Strains | Description | Growth in[a]: | |
|---|---|---|---|
| | | Atmospheric air | 5% CO$_2$ |
| *E. coli* | | | |
| Δ*can* | CA deletion mutant | − | + |
| Δ*can*(pRB473*can*-Sc) | Mutant complemented with CA from *S. carnosus* | + | + |
| Δ*can*(pRB473) | Mutant complemented with empty plasmid (control) | − | + |
| Δ*can*(pRB473*can*-Sp) | Mutant complemented with CA from *S. pseudintermedius* | + | + |
| *S. carnosus* | | | |
| Δ*can* | CA deletion mutant | − | + |
| Δ*can*(pRB473*can*-Sc) | Mutant complemented with its own CA | + | + |
| Δ*can*(pRB473) | Mutant complemented with empty plasmid (control) | − | + |
| *S. pseudintermedius* | | | |
| Δ*can* | CA deletion mutant | − | + |
| Δ*can*(pCtuf*can*-Sp) | Mutant complemented with its own CA | + | + |
| Δ*can*(pCtuf) | Mutant complemented with empty plasmid (control) | − | + |
| Δ*can*(pRB473*mpsABC*) | Mutant complemented with *mpsABC* from *S. aureus* | + | + |
| *S. aureus* Δ*mpsABC*(pCtuf*can*-Sp) | *mpsABC* deletion mutant complemented with CA from *S. pseudintermedius* | + | + |

[a]+ indicates growth on agar plates after overnight incubation under the respective conditions, while − indicates no growth.

feature of all $\beta$-class CAs (21). This feature facilitates the ligation of the zinc active site, with sulfur atoms of the two Cys residues as well as a nitrogen atom from His. Additionally, two residues numbered 40 and 42 were detected, which corresponds to the Asp/Arg dyad. The pair is important to ensure that water is the fourth ligand to zinc and contributes to the proton transfer step of catalysis (22).

To screen for the functional activity of CA, we utilized *E. coli* EDCM636, which is a *can* deletion mutant (*E. coli* Δ*can*) and therefore cannot grow in atmospheric air (19) but only grows under high-CO$_2$ (5%) conditions. This mutant was used as a recipient strain for complementation with staphylococcal *can* genes. The expression of the *can* gene from *S. carnosus* (*can*-Sc) or *S. pseudintermedius* (*can*-Sp) enabled it to grow under low-atmospheric-CO$_2$ conditions (Table 2), indicating that the probable CA in both strains is functional. Although the *E. coli* CA shows a protein identity of only 26% to the staphylococcal CAs, full complementation was achieved. *E. coli* Δ*can* strains harboring the respective empty plasmids were used as a negative control.

**The *can* gene from *S. carnosus* and *S. pseudintermedius* is required for growth under atmospheric CO$_2$.** In order to determine the importance of the *can* gene for staphylococcal growth, we constructed *can* deletion mutants (Δ*can*) in *S. carnosus* and *S. pseudintermedius*. The deletion of *can*-Sc in *S. carnosus* and *can*-Sp in *S. pseudintermedius* led to a severe growth defect (Fig. 2A and Table 2), a phenotype similar to those of the *E. coli* Δ*can* strain and also the *S. aureus* Δ*mpsABC* mutant and which requires a high concentration of CO$_2$ for growth. Complementation of these staphylococcal Δ*can* mutants reversed the phenotype, enabling their growth under atmospheric conditions without the need for high CO$_2$, indicating that the *can* gene is the only gene required for growth under atmospheric air conditions (Table 2). As negative controls, both of the deletion mutants carrying empty plasmids did not grow under atmospheric air conditions, whereas all of the strains grew under 5% CO$_2$.

***S. carnosus* and *S. pseudintermedius* Δ*can* mutants can be complemented by *S. aureus mpsABC* and vice versa.** Earlier we reported that MpsAB and CA represent a CO$_2$/bicarbonate-concentrating system in which each can functionally replace the other (14). Here, we wanted to address this hypothesis with reciprocal complementation within the staphylococcal species. We transformed both the *S. carnosus* Δ*can* and *S. pseudintermedius* Δ*can* mutants with plasmid pRB473*mpsABC*, containing the *S. aureus*-specific *mpsABC* genes. Both the Δ*can* strains could be complemented by *mpsABC* and grew like its parental counterparts under atmospheric conditions (Table 2). Likewise,

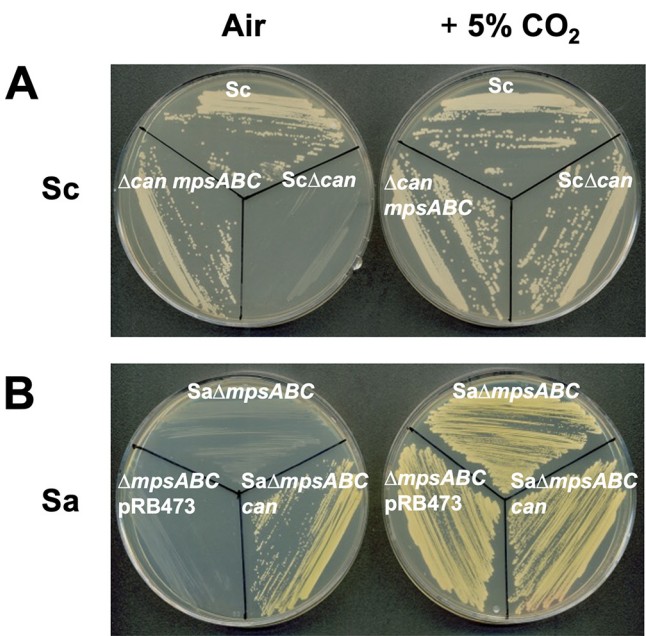

**FIG 2** The *S. carnosus* CA deletion mutant cannot grow in atmospheric air and is interchangeable with *mpsABC* from *S. aureus*. (A) The *S. carnosus* CA deletion mutant cannot grow in atmospheric air (0.04% $CO_2$) but can be restored at 5% $CO_2$ and can also be complemented with *mpsABC* from *S. aureus*. Clockwise from top: *S. carnosus* TM300 wild type (Sc), *S. carnosus* TM300 CA deletion mutant (Sc$\Delta$can), and Sc$\Delta$can complemented with plasmid pRB473 carrying *mpsABC*-Sa ($\Delta$can *mpsABC*). Likewise, CA from *S. carnosus* can also complement *S. aureus* $\Delta$*mpsABC*. (B) Clockwise from top: *S. aureus* HG001 $\Delta$*mpsABC* (Sa$\Delta$*mpsABC*), *S. aureus* HG001 $\Delta$*mpsABC* complemented with plasmid pRB473 carrying CA from *S. carnosus* (Sa$\Delta$*mpsABC* can), and *S. aureus* HG001 $\Delta$*mpsABC* carrying empty plasmid pRB473 as a control ($\Delta$*mpsABC* pRB473). Plates were incubated in atmospheric air (left) and 5% $CO_2$ (right).

reciprocal complementation showed that the *can* gene also rescued the growth of *S. aureus* $\Delta$*mpsABC* mutants under atmospheric air, indicating the interchangeable relationship between MpsAB and CA (Fig. 2B).

**MpsAB outperforms CA with regard to $CO_2$/bicarbonate concentration.** Given the interchangeable relationship of MpsAB and CA, we sought to examine if one system is superior to the other at concentrating $CO_2$/bicarbonate. We coexpressed MpsAB in *S. carnosus* and CA in *S. aureus* so that the strains have an extra $CO_2$/bicarbonate concentration system in addition to its own. The *S. aureus* $\Delta$*mpsABC* and *S. carnosus* $\Delta$*can* strains cannot grow at atmospheric $CO_2$ levels, suggesting that *mpsAB* and *can* are not only essential in their respective species, but also represent the sole $CO_2$/bicarbonate concentration system. When the *can* gene from *S. carnosus* was coexpressed in *S. aureus*(pRB473*can*-Sc), there was no evident increase in the growth compared to that of the wild type under atmospheric conditions (Fig. 3A). In contrast, the growth of the coexpressed strains seemed to decrease slightly compared to the wild type, and the decrease was more evident under $CO_2$ conditions. It appears that too much $CO_2$/bicarbonate attenuates the growth rather than enhances it. As no growth benefit was seen, we investigated the growth in the absence of MpsAB using the *S. aureus* $\Delta$*mpsABC* (pRB473*can*-Sc) reciprocal complemented strain. The growth of this strain was similar under both atmospheric and $CO_2$ conditions. The cloning of *can*-Sc could not fully complement the growth of the *S. aureus* $\Delta$*mpsABC* strain, but only to the levels of the deletion mutant grown under 5% $CO_2$ (shown by the arrow in Fig. 3A).

Likewise, the same observation was seen for the *S. carnosus* wild type (Fig. 3B). However, complementation of the *S. carnosus* $\Delta$*can* mutant with *mpsABC* or 5% $CO_2$ fully restored the growth of the *S. carnosus* $\Delta$*can* mutant to wild-type levels (shown by the arrow in Fig. 3B). This is in stark contrast to what we observed with the *S. aureus* $\Delta$*mpsABC* mutant. Unlike the reciprocal complemented *S. aureus* strain, the addition of

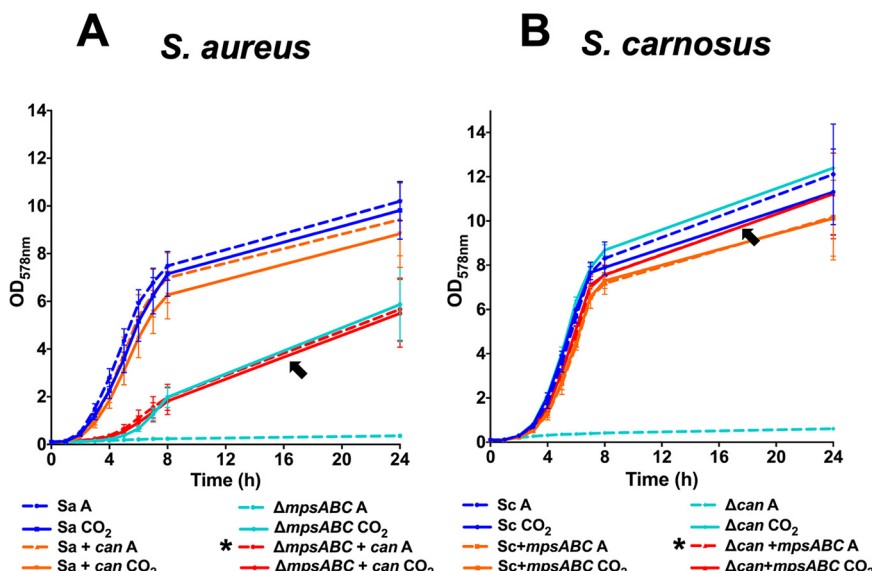

**FIG 3** Effect of coexpression of both MpsAB and CA on the growth of *S. aureus* and *S. carnosus*. The growth of *S. aureus* HG001 and *S. carnosus* TM300 strains under atmospheric air (A, dashed lines) and 5% $CO_2$ ($CO_2$, solid lines) conditions in 24 h. Coexpression of CA in *S. aureus* harboring MpsAB did not show any growth benefit, while coexpression of CA in *S. aureus* Δ*mpsABC* (marked with an asterisk in the figure key) could only partially complement the mutant to the same levels as in 5% $CO_2$, as shown by the arrow in panel A. (A) The growth of the *S. aureus* wild type (Sa), *S. aureus* complemented with CA from *S. carnosus* (Sa+*can*), *S. aureus* mpsABC deletion mutant (Δ*mpsABC*), and *S. aureus* mpsABC deletion mutant complemented with CA from *S. carnosus* (Δ*mpsABC*+*can*) under both atmospheric and $CO_2$ conditions. Likewise, coexpression of MpsAB in *S. carnosus* harboring CA did not show any growth benefit. However, the coexpression of MpsAB in *S. carnosus* Δ*can* (marked with an asterisk in the figure key) could fully complement the mutant to almost the wild-type levels and 5% $CO_2$, as shown by the arrow in panel B. (B) The growth of *S. carnosus* wild type (Sc), *S. carnosus* complemented with *mpsABC* from *S. aureus* (Sc+*mpsABC*), and the *S. carnosus* CA deletion mutant (Δ*can*), the *S. carnosus* can deletion mutant complemented with *mpsABC* from *S. aureus* (Δ*can*+*mpsABC*) under both conditions. Each point in the graph is the mean ± standard deviation (SD) from three independent biological replicates.

*mpsABC* could restore the growth of the *S. carnosus* Δ*can* mutant, suggesting that MpsAB outperforms CA with regard to $CO_2$/bicarbonate concentration, as reflected by the higher growth under low-atmospheric-$CO_2$ conditions. This observation led us to the question of why the growth of the *S. aureus* Δ*mpsABC* mutant can be fully restored neither by cloning of *can* nor by 5% $CO_2$. Therefore, we hypothesized that in contrast to *S. carnosus*, the diffusion of $CO_2$ in *S. aureus* is hindered.

**Does biofilm hinder the diffusion of $CO_2$ into the cell?** Compared to *S. carnosus*, *S. aureus* and *S. epidermidis* produce polysaccharide intercellular adhesin (PIA), which is composed of linear β-1,6-linked glucosaminylglycans (23–25). PIA is of slimy consistency, loosely surrounding the cells, and is the basis for tight biofilm formation (26). Therefore, PIA could have a negative effect on the diffusion of $CO_2$ into the cell. Following on to this, we searched for the occurrence of MpsAB, CA, and the intercellular adhesion gene (*ica*)-encoded protein cluster, which mediates the formation of biofilm in selected staphylococcal species. Table 3 shows the presence of such genes/proteins based on Pfam domains of each *ica* gene. Most of the strains can be divided into four categories of having MpsAB and Ica (group i), CA only (group ii), MpsAB only (group iii), and CA and Ica (group iv). The first two categories, groups i and ii, support our hypothesis that on the one hand, MpsAB and Ica are correlated, and on the other hand, CA predominates in those species lacking *ica* genes. However, groups iii and iv are exceptions. To provide more evidence for the hypothesis, we examined *S. aureus* Δ*ica* mutants and the expression of *icaADBC* genes in *S. carnosus*.

**Deletion of the *ica* operon in the *S. aureus* Δ*mpsABC* Δ*ica* mutant increased growth in the presence of $CO_2$.** In *S. aureus*, we deleted the *ica* operon, including its regulator in the background of the *S. aureus* Δ*mpsABC* mutant. Indeed, under $CO_2$

**TABLE 3** Occurrence of MpsAB, CA, and intercellular adhesion gene (*ica*)-encoded protein cluster in selected staphylococcal species

| Protein(s) and species | Presence of MpsAB, CA, or Ica protein[a] | | | | |
|---|---|---|---|---|---|
| | MpsAB | CA | IcaA | IcaB | IcaC |
| **Group i: MpsAB and Ica** | | | | | |
| *Staphylococcus argenteus* BN75 | + | − | + | + | + |
| *Staphylococcus aureus* subsp. *aureus* MSHR1132 | + | − | + | + | + |
| *Staphylococcus aureus* subsp. *aureus* USA300_FPR3757 | + | − | + | + | + |
| *Staphylococcus capitis* AYP1020 | + | − | + | + | + |
| *Staphylococcus epidermidis* RP62A | + | − | + | + | + |
| *Staphylococcus lugdunensis* C_33 | + | − | + | + | + |
| *Staphylococcus simiae* NCTC 13838 | + | − | + | + | + |
| *Staphylococcus xylosus* SMQ121 | + | − | + | + | + |
| *Staphylococcus saprophyticus* 883 | + | − | ++ | ++ | ++[b] |
| *Staphylococcus cohnii* SNUDS-2 | + | − | + | − | + |
| *Staphylococcus sciuri* SNUSD-18 | (+) | + | ++ | − | ++ |
| **Group ii: CA only** | | | | | |
| *Staphylococcus agnetis* 908 | − | + | − | − | − |
| *Staphylococcus carnosus* TM300 | − | + | − | − | − |
| *Staphylococcus felis* ATCC 49168 | − | + | − | − | − |
| *Staphylococcus hyicus* ATCC 11249 | − | + | − | − | − |
| *Staphylococcus lutrae* ATCC 700373 | − | + | − | − | − |
| *Staphylococcus muscae* NCTC 13833 | − | + | − | − | − |
| *Staphylococcus schleiferi* 1360-13 | − | + | − | − | − |
| **Group iii: MpsAB only** | | | | | |
| *Staphylococcus equorum* KS1039 | + | − | − | − | − |
| *Staphylococcus haemolyticus* JCSC1435 | + | − | − | − | − |
| *Staphylococcus hominis* subsp. *hominis* K1 | + | − | − | − | − |
| *Staphylococcus nepalensis* JS1 | + | − | − | − | − |
| *Staphylococcus pasteuri* SP1 | + | − | − | − | − |
| *Staphylococcus succinus* 14BME20 | + | − | − | − | − |
| *Staphylococcus warneri* SG1 | + | − | − | − | − |
| **Group iv: CA and Ica** | | | | | |
| *Staphylococcus condimenti* DSM 11674 | − | + | + | + | + |
| *Staphylococcus pettenkoferi* FDAARGOS_288 | − | + | +[c] | + | + |
| *Staphylococcus piscifermentans* NCTC 13836 | − | + | + | + | + |
| *Staphylococcus pseudintermedius* ED99 | − | + | + | + | + |
| *Staphylococcus stepanovicii* NCTC 13839 | − | + | + | + | + |
| *Staphylococcus simulans* FDAARGOS_124 | − | + | + | − | + |

[a]The presence of the proteins was inferred based on the following Pfam domain searches from finished bacterial genomes in the Integrated Microbial Genomes & Microbiomes (IGM/G) database: MpsAB was from Pfam00361 and PFam10070, respectively, carbonic anhydrase (CA) was based on prokaryotic type-carbonic anhydrase (Pfam00484) and eukaryotic-type CA (Pfam00194), and the biofilm-associated proteins are based on the Ica-encoding genes *icaA*, which has the domain glycosyl transferase family 2 (Pfam00535), *icaB*, which contains the domain polysaccharide deacetylase (Pfam01522), and *icaC*, which has the domain acyltransferase family (Pfam01757). The symbols + and − indicate the presence or absence, respectively, of the protein domains. The symbol +/− indicates the presence or absence of the protein domains, with + indicates one and ++ indicate two protein domains. The symbol (+) indicates that in *S. sciuri* SNUSD-18, MpsA and MpsB appear to be truncated.
[b]Two sets of operons of *ica* genes in different locations were detected in the genome of *S. saprophyticus* 883.
[c]*icaA* was found to contain Pfam13641 which is annotated as glycosyltransferase-like family 2, but the protein has 76% similarity (BLASTp) to the IcaA from *S. aureus*.

conditions, the *S. aureus* Δ*mpsABC* Δ*ica* mutant grew to a higher optical density (OD) than the Δ*mpsABC* mutant, although not quite to the wild-type levels (indicated by the black arrow in Fig. 4A).

**Expression of the *ica* operon in the *S. carnosus* Δ*can* mutant decreases growth in the presence of $CO_2$.** To confirm the effect of PIA, we transformed plasmid pTX30*icaADBC*, which encodes PIA biosynthesis into the *S. carnosus* Δ*can* mutant. In this plasmid, the *ica* expression is xylose inducible. Under inducing conditions and in the presence of 5% $CO_2$, the growth of the *S. carnosus* Δ*can*(pTX30*icaADBC*) strain was significantly reduced, constituting only half of the *S. carnosus* Δ*can* mutant grown with $CO_2$ and the wild type (indicated by the black arrow in Fig. 4B).

As controls, the addition of xylose and the presence of the empty plasmid (pTX30) did not influence the growth of the *S. carnosus* Δ*can* mutant or wild type under both atmospheric and $CO_2$ conditions (see Fig. S2A, B, D, and E in the supplemental material). In addition, induction of PIA expression in wild-type *S. carnosus*(pTX30*icaADBC*)

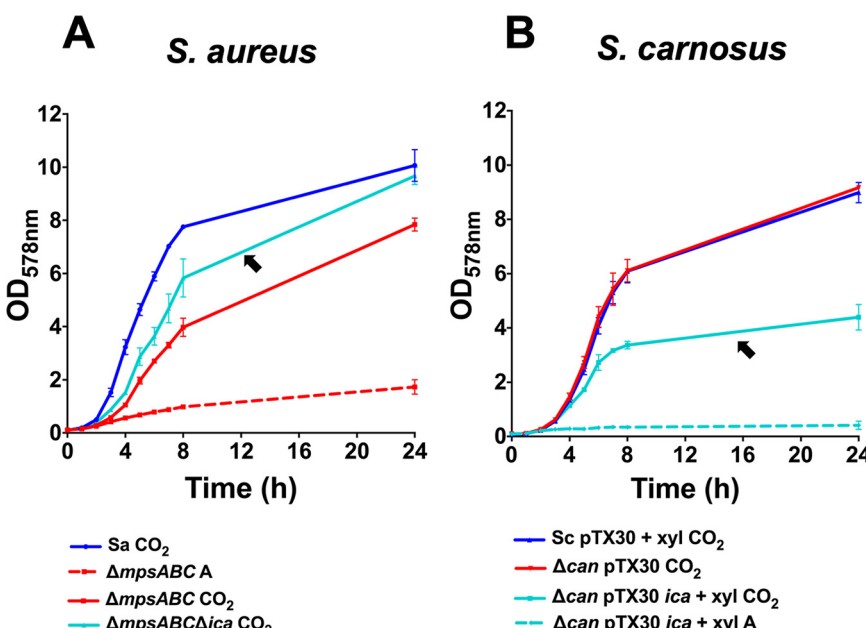

**FIG 4** The influence of biofilm mediated by polysaccharide intercellular adhesin (PIA) encoded by the *ica* operon in the growth of staphylococcal strains. (A) The growth of the *S. aureus* HG001 wild type (Sa), HG001 *mpsABC* deletion mutant (Δ*mpsABC*), and HG001 *mpsABC* and *ica* deletion mutant (Δ*mpsABC ica*) under atmospheric (A, dashed lines) and 5% $CO_2$ ($CO_2$, solid lines) conditions for 24 h. The arrow shows the Δ*mpsABC* Δ*ica* double mutant in which the biofilm-associated genes were deleted and which grew to a higher OD than the Δ*mpsABC* mutant alone in $CO_2$. (B) The growth of the *S. carnosus* TM300 wild type (Sc) and TM300 carbonic anhydrase gene deletion mutant (Δ*can*) carrying empty plasmid pTX30 (as a control) under atmospheric (A, dashed lines) and 5% $CO_2$ ($CO_2$, solid lines) conditions for 24 h. The arrow shows the Δ*can*(pTX30) strain carrying the *ica* genes required for biofilm formation, which grew to a much lower OD than the Δ*can* strain alone in $CO_2$ when induced by 0.7% xylose (xyl). All growth studies were performed using TSB. Each point in the graph is the mean ± SD from three independent biological replicates.

did not affect growth (Fig. S2C). Taken together, these observations support our hypothesis that PIA formation hinders the diffusion of $CO_2$ into the cells with pronounced reduction of growth in the *S. carnosus* Δ*can* mutant.

**Coexpression of both MpsAB and CA does not confer any growth benefits under stress conditions.** Next, we proceeded to find out if having both the $CO_2$ and bicarbonate concentration systems confers any advantage under different environmental stress. To confirm a potential growth benefit in all backgrounds, we used three strains of *S. aureus* and *S. carnosus,* respectively, which included the wild type, strains coexpressing both systems, and reciprocal complemented strains. Cells were exposed to different concentrations of NaCl ranging from 0% to 7.5% (see Fig. S3 in the supplemental material), to pHs from 5.5 to 8.5 (see Fig. S4 in the supplemental material), or to iron limitation by adding the iron chelator 2,2-dipyridyl (DIP) (see Fig. S5 in the supplemental material). We also tested the effect of temperature at 25, 37, and 42°C (see Fig. S6 in the supplemental material). However, we observed no growth benefit when both systems were present under any of the applied stress conditions. Collectively, these results indicate that the coexpression of MpsAB and CA did not confer any growth benefits under the stress conditions tested.

**Phylogenetic analysis of CA.** The large number of strains listed in Table S1 necessitated further phylogenetic analysis to explore the relationship regarding the distribution of CA in *Firmicutes*. Since we confirmed in this study that the *S. carnosus* CA is functional, we used the protein sequence as a query sequence to search for homologs in NCBI database, and the resulting sequences were used to construct a phylogenetic tree. The CA homologs are widespread in the phylum *Firmicutes,* with a high degree of conservation among *Staphylococcaceae* (see Fig. S7 in the supplemental material). Among the main genera seen are *Planomicrobium, Bacillus, Clostridium, Streptococcus,*

and *Oceanobacillus*, with *Lactobacillus delbrueckii* being the most distantly related to the CA protein of *S. carnosus*. Some *Streptococcus* species, such as *Str. parauberis*, *Str. didelphis*, and *Str. suis*, form a distinct clade from the rest of the *Firmicutes*. A closer look at the family *Staphylococcaceae* revealed that CA is further split into two subclades: the first consists of *Staphylococcus* and *Macrococcus* and the second of *Jeotgalicoccus* and *Salinicoccus* (Fig. S7B). With respect to the *S. carnosus* CA, the most closely related species is *S. simulans*, whereas the most distantly related species is *Auricoccus indicus*. Of note, a few of the *Staphylococcus* species, *S. stepanovicii*, *S. lentus*, and *S. sciuri*, split and form a separate clade from the others in the same genus.

**All completely sequenced *S. aureus* genomes contain only the *mpsAB* operon but no *can* gene.** According to the IMG/G database, there are 209 finished genomes of *S. aureus* (accessed 20 October 2020). We checked all of these genomes for the presence of putative CA homologs to *S. carnosus*, but none was found in any of these genomes. On the basis of the Pfam motif of CA, no potential CA homologs was present. In addition, a search in all these genomes for any protein annotated as CA also yielded no result. Furthermore, we did a BLAST search of the *S. carnosus* $\beta$-CA against the NCBI database and found two protein sequences from an *S. aureus* strain with low similarity to *S. carnosus* CA. However, the sequences were derived from an unfinished genome (permanent draft) and may not be properly sequenced/annotated. In addition, we also did not find significant homology (BLASTp) with $\alpha$-CAs from *Enterococcus faecium*, *Helicobacter pylori*, *Neisseria gonorrhoeae*, and *Vibrio cholerae* and human CA1 and CA2, as well as $\gamma$-CAs from *Enterococcus faecium*, *Escherichia coli*, *Halobacterium salinarum*, and *Methanosarcina thermophila*. Given the extensive search performed, it is extremely unlikely that CA is present in *S. aureus*.

## DISCUSSION

In the present study, we have followed up on our previous work on the interchangeable roles of MpsAB and CA in $CO_2$/bicarbonate-concentrating systems and described a previously unreported CA activity in *Staphylococcus*. *Firmicutes* harboring MpsAB or CA have adapted to diverse habitats from soil, animals, and plants to deep ocean (Table 1; Table S1). However, the ecological and metabolic diversity of these *Firmicutes* renders it difficult to group them based solely on the presence of MpsAB and CA. A first look at the distribution of MpsAB and CA within *Firmicutes* families shows a seemingly arbitrary distribution in the respective species: some have only MpsAB or CA, some have both, and others have two or more genes for $\beta$- or $\alpha$-CA. At the moment, we have no clear explanation on the distribution of MpsAB and CA among the members of the *Firmicutes*.

Phylogenetic analyses of CA in *Firmicutes*, particularly in *Staphylococcaceae*, revealed that only related genera and species are clustered. However, no conclusion could be made about the evolution of CAs since we do not know the common ancestor of these phyla. On another note, there are only some bacteria in which the $\beta$-CA activity was thoroughly described to date (3) such as in *E. coli* (19), *Streptococcus pneumoniae* (18), or *Clostridium autoethanogenum* (27). In other classes, the $\alpha$-CA includes studies in *Helicobacter pylori* (28), *Neisseria gonorrhoeae* (29), *Mesorhizobium loti* (30), and *Rhodopseudomonas palustris* (31), while $\gamma$-CA was well characterized in *Methanosarcina thermophila* (32). Different classes of CAs ($\alpha$ and $\beta$) have been reported in the same bacteria, such as *Thiomicrospira crunogena* (33). To the best of our knowledge, no credible study has investigated the activity and role of CA in staphylococci. For this reason, we selected *S. carnosus* and *S. pseudintermedius* for a more detailed study. *S. carnosus* is a nonpathogenic food-grade *staphylococcus* strain that does not possess *ica* genes and therefore does not produce biofilm, while *S. pseudintermedius* is a canine pathogen carrying the *ica* genes (Table 3). Both species possess only one CA gene (*can*), and when this gene was deleted, the mutants were not able to grow in ambient air. Such a phenotype was also seen in other CA mutant strains of bacteria like *E. coli* and *Str. pneumoniae*, further emphasizing the importance of $CO_2$/bicarbonate concentration systems.

Given that some genes have no detectable fitness effects under normal conditions and only exert their effects under certain stress situations (34), we repeated the experiments under various environmental stress stimuli, but the coexpression of both systems did not confer any significant growth benefits under these conditions (Fig. S3 to S6). As with all growth studies shown here (Fig. 3; Fig. S3), we observed that the presence of both MpsAB and CA did not improve growth but instead slightly reduced it, particularly under $CO_2$ conditions. In principle, it does not really make sense if bacteria have both an $HCO_3^-$ transporter and a CA because the $HCO_3^-$ transported into the cell will be converted by CA to $CO_2$, which can escape the cells by diffusion. Thus, the benefit of a transporter is mitigated by the presence of a CA. This is exactly what we observed experimentally by coexpressing *can* in *S. aureus*(pRB473*can*-Sc), which caused decreased growth compared to the wild type. Indeed, most of the bacterial species have only one or the other system. Nevertheless, there are a few species which have both systems, like some endospore-forming bacilli and clostridia (Table 1). In such cases where microorganisms undergo morphological differentiation, it might be advantageous to have both systems, where both systems are active in different cell compartments.

An interesting observation in this study is that the growth of the *S. aureus* Δ*mpsABC* mutant could only be partially complemented by *can* or 5% $CO_2$, while the growth of the *S. carnosus* Δ*can* mutant complemented with *mpsABC* or 5% $CO_2$ could be almost fully restored (Fig. 3). One possible explanation for this finding could be that $CO_2$ diffuses less easily into the cell in *S. aureus* than in *S. carnosus*. While bacteria with CA depend on the diffusion of $CO_2$, MpsAB-harboring bacteria are independent of $CO_2$ diffusion because they transport bicarbonate from outside into the cell. In this context, we hypothesized that in *S. aureus*, the diffusion of $CO_2$ is impaired, and therefore the $CO_2$ levels entering the cells are insufficient to fully complement the growth in the *S. aureus* Δ*mpsABC* mutant by either 5% $CO_2$ or cloning of *can*. We assumed that the diffusion barriers are found in the cell envelope, and its major differences between *S. aureus* and *S. carnosus* may affect $CO_2$ diffusion in *S. aureus*.

The *S. aureus* cell wall is much richer in cell-wall-anchored adhesin proteins such as protein A, fibronectin, and fibrinogen or collagen binding proteins (35), which are not present in *S. carnosus* (16, 36, 37). Another important difference between the two species is that *S. aureus* has mucus substances attached to its cell surface, which are not present in *S. carnosus*. One of the mucus substances is PIA (26), which is encoded by the *ica* genes (23–25). The abundance of adhesins and the mucilage could possibly impair the diffusion of $CO_2$ in *S. aureus*. Moreover, most *S. aureus* strains produce type 5 and 8 capsular polysaccharides (38, 39), which might further contribute to the impairment.

The first two groups in Table 3 were indeed consistent with our hypothesis. For group i, it seems to be advantageous to have a bicarbonate transporter (MpsAB) instead of CA since the PIA produced by these species form the basis of biofilm. For group ii, CA appears to be fully adequate since there is no PIA to interfere with the $CO_2$ diffusion. Only groups iii and iv do not seem to agree with our assumption. The lack of a strict correlation between the last two groups could be attributed to acquisition of *mpsAB* or *ica* by horizontal gene transfer. Furthermore, we have no indication if the *ica* genes are expressed at all in this group. Another possible explanation may be the interaction of the microbial community in the natural environment. Some bacteria that rely on high levels of $CO_2$ but lack CA can still grow in commensal situations in this context (40).

Deletion of *ica* genes in the *S. aureus* Δ*mpsABC* mutant, although not fully, improved the growth in 5% $CO_2$. Incomplete restoration could be due to the capsule polymers that are still present. Conversely, the cloning of *ica* genes into the *S. carnosus* Δ*can* mutant decreased its growth as this strain cannot be fully supplied with $CO_2$ in the presence of PIA. We are aware that the *ica* expression could lead to cell aggregation, which could falsify the OD measurements. Therefore, we always vortexed the samples rigorously, and appropriate controls were made by using all the clones as shown in Fig. S2. Clearly, these

are only indirect results. However, it should be noted that a direct measurement of $CO_2$ diffusion in aqueous medium is almost impossible due to its high dissociation capability. Nevertheless, biophysical calculations and measurements suggest that gas diffusion is impaired in the biofilm mode of growth due to sorption to the biofilm matrix (41, 42).

Recently we showed that MpsAB homologs are widespread in the *Firmicutes* phylum (14). Among them are pathogens such as *Bacillus anthracis*, possessing capsule and S-layer proteins, *Vibrio cholerae*, harboring a capsular layer, or *Legionella pneumophila*, which has its S-layer proteins. All of these pathogens are distinguished by a surface mucilage that may retard $CO_2$ diffusion and thus making a bicarbonate transporter more advantageous than CA. Other pathogens like *Streptococcus pyogenes*, *Enterococcus faecalis*, *Mycobacterium tuberculosis*, or *E. coli* have only CA homologs (14). Interestingly, the lipid bilayer of *M. tuberculosis* should also retard $CO_2$ diffusion; however, this might be compensated for by possessing three CA homologs (14).

Finally, the aforementioned findings and that >200 fully sequenced *S. aureus* genomes harbor only MpsAB and no CA diminish the possibility that CA is present in *S. aureus*. These findings are in contradiction with other publications about the presence of CA in *S. aureus*, which must be reconsidered. In conclusion, we underlined the importance of $CO_2$/bicarbonate-concentrating systems in *Firmicutes* that depend on either MpsAB or CA. Despite the apparent lack of growth benefits when both are coexpressed in the same strain, we found that MpsAB outperforms CA on the basis of growth restoration. Our findings suggest that in certain species, the $CO_2$ diffusion is hindered by mucilage, slime, capsules, or other polymers. For these species, it is advantageous if they have an MpsAB-bicarbonate transporter. With such a transporter, the cells are not dependent on $CO_2$ diffusion for their carboxylation reactions (Fig. 5). Against this background, it is thus conceivable that *S. aureus* and other bacterial species have a membrane-localized bicarbonate/$CO_2$ transporter instead of a cytoplasmic CA.

## MATERIALS AND METHODS

**Bacterial strains and growth conditions.** All strains used in this study are listed in Table S2 in the supplemental material. For cloning procedures, all *E. coli* and staphylococcal strains were cultivated in at 37°C with shaking at 150 rpm in basic medium (BM), unless otherwise specified. The BM consists of 1% soy peptone, 0.5% yeast extract, 0.5% NaCl, 0.1% glucose, and 0.1% $K_2HPO_4$ at pH 7.2. All cultures were grown in 10 ml medium using baffled 100-ml shake flasks, except for growth studies, in which cells were grown in 15 ml medium. When necessary, the culture medium was supplemented with the following antibiotics used at the indicated concentrations: 10 $\mu$g/ml chloramphenicol for staphylococcal strains and 100 $\mu$g/ml ampicillin and 30 $\mu$g/ml kanamycin for *E. coli* strains.

**Construction of staphylococcal deletion mutants.** All oligonucleotides used in this study are listed in Table S3 in the supplemental material. The nucleotide and amino acid sequences were obtained from the Kyoto Encyclopedia of Genes and Genomes (KEGG). The *S. carnosus* TM300 Δ*can* (KEGG accession no. SCA_1457) and *S. pseudintermedius* ED99 Δ*can* (KEGG accession no. SPSE_0869) deletion mutants were constructed as markerless deletions using allelic replacements as described by Bae and Schneewind (43). Briefly, approximately 1-kb fragments of upstream and downstream of the carbonic anhydrase-encoding gene (*can*) were amplified from the chromosomal DNA of *S. carnosus* TM300 and *S. pseudintermedius* ED99, respectively. The amplified fragments were assembled with linearized plasmid pBASE6 (EcoRV restriction site) (44) via Gibson assembly (45) using Hi-Fi DNA assembly master mix (New England Biolabs). The resulting plasmid was transformed into chemically competent *E. coli* DC10B (46). The respective clones harboring the right genes were then transformed into *S. carnosus* TM300 via electroporation and into *S. pseudintermedius* ED99 by protoplast transformation (47). Deletion of the *can* genes in both of the strains was confirmed by PCR and sequence analysis.

For the construction of *S. aureus* HG001 Δ*mpsABC*, the *ica* operon genes consisting of the transcriptional regulator, *icaR*, as well as *icaA*, *icaD*, *icaB*, and *icaC* (corresponding to KEGG accession no. SAOUHSC_03001, SAOUHSC_03002, SAOUHSC_03003, SAOUHSC_03004, and SAOUHSC_03005, respectively) were deleted. Approximately 1,500-kb fragments upstream and downstream of the Ica operon-encoding genes were amplified from the chromosomal DNA of *S. aureus* HG001, and the amplified fragments were assembled with linearized plasmid pBASE6 (SmaI restriction site) via Gibson assembly. The remaining steps were performed as described above.

**Construction of complementation vectors.** The complementation for *S. carnosus* CA (*can*-Sc) was carried out using plasmid pRB473 (48). Along with its putative native promoter, the CA-encoding gene (*can*-Sc) was amplified from the chromosomal DNA of *S. carnosus* TM300 before being assembled into linearized pRB473 (SmaI restriction site) by Gibson assembly. The putative promoter regions were determined by DNA sequence analysis. As for the construction of complementation of *S. pseudintermedius* ED99 CA (*can*-Sp), the plasmid pCtufamp (49) was used. The CA-encoding gene (*can*) was amplified from the chromosomal DNA of *S. pseudintermedius* ED99 and then assembled into linearized pCtufamp

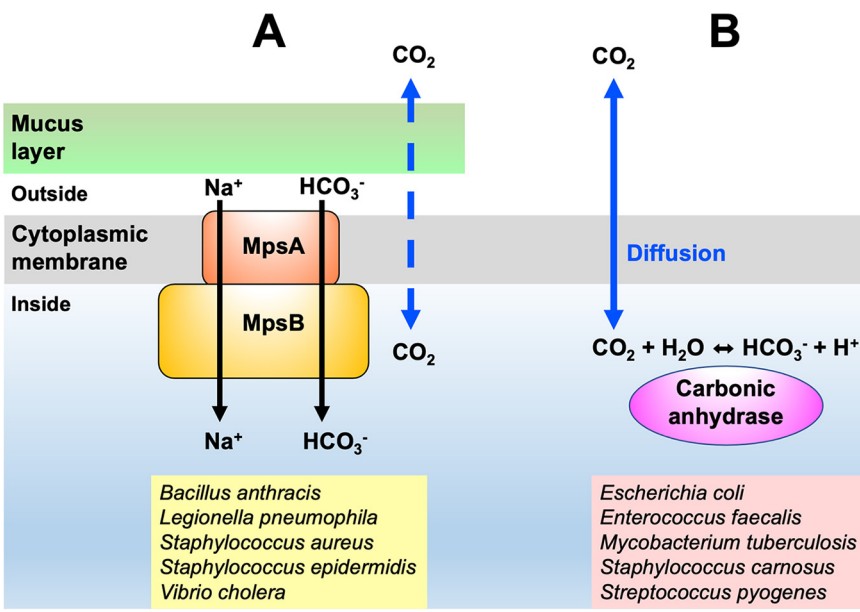

**FIG 5** Model of the restricted $CO_2$ diffusion in mucoid bacteria and the advantage of having a bicarbonate transporter. (A and B) Biofilm-producing bacteria such as *S. aureus* and *S. epidermidis* (A) utilize a membrane-localized transporter (MpsAB) to facilitate the movement of membrane-impermeant $HCO_3^-$ across the cell membrane, as well as (B) non-biofilm-producing bacteria such as *S. carnosus* and *S. felis* (B), whose cytoplasmic carbonic anhydrase (CA) activity is dependent on good $CO_2$ diffusion into the cell. The dashed blue arrow represents restricted $CO_2$ diffusion, and the solid blue arrow indicates unrestricted $CO_2$ diffusion. Additional examples of bacterial species in the MpsAB and CA group are listed at the bottom of the figure.

(PacI and HindIII restriction sites) by Gibson assembly. The constructed plasmids were first introduced into *E. coli* DC10B and then into the *S. carnosus* TM300 Δ*can* and *S. pseudintermedius* ED99 Δ*can* strains accordingly. Both of the recombinant plasmids (pRB473*can*-Sc and pCtuf*can*-Sp) were also introduced into *S. aureus* RN4220 before being transformed into the *S. aureus* HG001 wild type and *S. aureus* Δ*mpsABC* mutant (14). Additionally, both of the recombinant plasmids were also transformed into EDCM636 (*E. coli* Δ*can*) which is an *E. coli* MG1655 derivative harboring a kanamycin resistance marker replacing a deletion of the carbonic anhydrase-encoding gene *can* (19). The strain was purchased from *E. coli* Genetic Stock Center, Yale University. All of the staphylococcal strains and the *E. coli* Δ*can* mutant were also transformed with the empty plasmids pRB473 and pCtufamp as controls.

For the complementation of *S. carnosus* TM300 Δ*can* with *ica* genes, plasmid pTX30 containing the entire *ica* operon (*icaADBC*) under the control of a xylose-inducible promoter (24) was used and transformed into *S. carnosus* TM300 Δ*can*. As negative controls, the empty plasmid pTX30 was transformed into *S. carnosus* TM300 and *S. carnosus* TM300 Δ*can*.

**Reciprocal complementation.** To investigate if the MpsAB and CA could complement reciprocally, plasmid pRB473 carrying *mpsABC* from *S. aureus* HG001(pRB473*mpsABC*) from our previous study (14) was used for transformation into the *S. carnosus* TM300 Δ*can* and *S. pseudintermedius* ED99 Δ*can*, respectively, via protoplast transformation.

**Growth and complementation studies.** Tryptic soy broth (TSB) (Sigma-Aldrich) and tryptic soy agar (TSA) were used for growth studies involving staphylococcal strains. For growth characterization on solid medium, all staphylococcal strains were cultured for 24 h under atmospheric conditions and continuous shaking at 37°C, except for the *S. aureus* Δ*mpsABC* complemented strain, which was precultured in 5% $CO_2$ due to its slow growth. The 5% $CO_2$ condition was achieved in a $CO_2$ incubator (Heraeus Instruments). The cultures were subsequently adjusted to an $OD_{578}$ of 5 and then streaked on agar plates and incubated overnight in atmospheric and 5% $CO_2$ conditions. Images of the agar plates were taken using an image scanner (Epson). The complementation experiments of *E. coli* EDCM636 were also performed as described above, except that BM agar was used instead of TSA.

For growth characterization of staphylococcal strains in liquid medium, the strains were precultured in TSB as described above. The main cultures were inoculated to an $OD_{578}$ of 0.1 and grown under atmospheric and 5% $CO_2$ conditions under continuous shaking. Aliquots were taken for $OD_{578}$ measurements at 0 h and every 1 h until 8 h and then at 24 h. Growth studies were performed in three independent biological replicates. For growth studies of *S. carnosus* involving plasmid pTX30, xylose at a final concentration of 0.7% was added to the TSB. For bacterial cultures with the presence of pTX30 *ica*, the flasks were vortexed vigorously before the OD measurements were taken to disrupt clumps due to possible biofilm formation.

**Occurrence of CA and MpsAB homologs and Ica based on Pfam domains.** The occurrence of CA and MpsAB protein homologs was inferred from a Pfam domain (50) search from finished bacterial

genomes found in the Integrated Microbial Genomes & Microbiomes (IGM/G) database (accessed 1 September 2020) (15). MpsA and MpsB belong to Pfam00361 and Pfam10070, respectively, while Pfam00484 and Pfam00194 are members of the prokaryotic and eukaryotic-type CAs, respectively. Other Pfam domains, such as Pfam08936 for carboxysome shell carbonic anhydrase (CsoSCA), Pfam18484 for cadmium CA repeat, and Pfam10563 for a putative CA-like domain, were also searched within the *Firmicutes* species, but yielded no results. Results are shown as the presence or absence of the respective Pfam domains and the frequency in a particular species if the domain is present.

For Ica-encoding genes, *icaA* belongs to Pfam00535, which has the domain glycosyl transferase family 2, while *icaB* and *icaC* are members of the domains Pfam01522 (polysaccharide deacetylase) and Pfam01757 (acyltransferase family), respectively. Results are presented as the presence and absence of the related Pfam domains and were divided into four categories.

**Phylogenetic trees.** Homologs of the *S. carnosus* TM300 CA (NCBI Protein ID accession no. CAL28362) were identified from the NCBI database (51) using Protein BLAST (accessed 17 September 2020). Searches were conducted within the bacterial *Firmicutes* phylum and bacterial family *Staphylococcaceae*. Selected protein sequences of mostly one strain per species in the *Firmicutes* phylum and *Staphylococcaceae* family were aligned using ClustalW (52). The multiple sequence alignments comprising 103 taxa in *Firmicutes* and 50 taxa in *Staphylococcaceae*, respectively. Phylogenetic trees were constructed using the maximum likelihood method and JTT matrix-based model (53). Results were assessed using 500 bootstrap replicates conducted in MEGAX (54, 55). The trees with the highest log likelihood ($-18,048.25$ for *Firmicutes* and $-6,526.83$ for *Staphylococcaceae*) are shown. The percentage of trees in which the associated taxa clustered together is displayed next to the branches. Initial trees for the heuristic search were obtained automatically by applying Neighbor-Join and BioNJ algorithms to a matrix of pairwise distances estimated using the JTT model and then selecting the topology with superior log likelihood value. The tree is drawn to scale, with branch lengths measured in the number of substitutions per site. The analyses involved 103 and 50 amino acid sequences, with a total of 208 and 194 positions in the final data set for *Firmicutes* and *Staphylococcaceae*, respectively.

**Growth studies for stress tolerance.** To evaluate if MspAB or CA confers any benefits under environmental stress, we exposed the bacterial cells to different stresses using NaCl, pH, iron limitation, and temperature and measured the growth in terms of $OD_{578}$ using a multiplate reader (Varioskan Lux; Thermo Scientific). For all stress conditions, six bacterial strains were used: the *S. aureus* HG001 wild type, *S. aureus* HG001(pRB473*can*-Sc), *S. aureus* HG001 Δ*mpsABC* complemented with pRB473*can*-Sc, *S. carnosus* TM300 wild type, *S. carnosus* TM300(pRB473*mpsABC*) and *S. carnosus* TM300 Δ*can* complemented with pRB473*mpsABC*. For salt stress tolerance studies, cells were precultured in LB without NaCl. Overnight cultures were washed once with LB (without NaCl) before being resuspended in the same medium. NaCl was added to LB with final concentrations of 2.5, 5, and 7.5% and with no NaCl (0%) as a negative control. Cells were added to the LB medium at a final OD of 0.01 in a total volume of 500 $\mu$l each in a 48-well microplate. The microplate was sealed with Parafilm to prevent evaporation and incubated at 37°C with shaking in a multiplate reader. OD readings were recorded every 30 min for 24 h. For pH stress, cells were precultured in LB medium at pH 7.2, and overnight cultures were washed once with the LB before being resuspended in the same medium. Washed cells were added to LB medium adjusted to pHs 5.5, 6.5, 7.5, and 8.5, and the plate was incubated as with the same protocols as NaCl studies. For iron limitation studies, cells were precultured in TSB, and overnight cultures were washed and resuspended in TSB. The iron chelator 2,2-dipyridyl (DIP) was dissolved in ethanol and added to TSB at final concentrations of 250 $\mu$M, 500 $\mu$M, 1 mM, and 2 mM. Cells at a final OD of 0.01 were added to the TSB, and growth studies were carried out as with the NaCl studies. All growth studies were performed in three independent biological replicates. For temperature stress, overnight cultures of the cells were adjusted to an OD of 0.5 and then streaked on LB agar plates prior to incubation at room temperature, 37°C, and 42°C for 24 h.

**Data visualization.** Multiple-sequence alignment of the protein sequences in Fig. 2 were obtained from IMG/M database and aligned using Clustal Omega (56). The percentages of identity of the CA protein from *S. carnosus* to the other sequences were compared using BLASTp (51). The growth curves were visualized using GraphPad Prism 6.0 software.

**Data availability statement.** The main data supporting the findings of this work are available within the article and its supplemental material files or from the corresponding author upon request.

## SUPPLEMENTAL MATERIAL

Supplemental material is available online only.

**SUPPLEMENTAL FILE 1**, PDF file, 1.9 MB.

## ACKNOWLEDGMENTS

This work was supported by funding from the Deutsche Forschungsgemeinschaft the Germany's Excellence Strategy—EXC 2124—390838134 "Controlling Microbes to Fight Infections" and the Ministry for Science, Research and the Arts of Baden-Wuerttemberg (MWK) "AntibioPPAP." S.-H.F. received a Ph.D. fellowship from the German Academic Exchange Service (DAAD). L.H. was supported by the Chinese Scholarship Council. We acknowledge support by the Open Access Publishing Fund of the University of Tübingen.

F.G. and S.-H.F. conceived the idea and designed the study. S.-H.F. performed most of the experiments. M.M., L.H., and P.M.T. performed some of the cloning experiments.

F.G. and S.-H.F. analyzed the data and wrote the manuscript. All authors read and approved the final manuscript.

We declare no conflicts of interest.

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
