## [Reviewer comments · Microbiology Spectrum]

**Microbiology
Spectrum**

The MpsAB bicarbonate transporter is superior to carbonic anhydrase in biofilm-forming bacteria with limited CO₂ diffusion

Sook-Ha Fan, Miki Matsuo, Li Huang, Paula Tribelli, and Friedrich Götz

Corresponding Author(s): Friedrich Götz, University of Tuebingen

Review Timeline:

Submission Date:	May 15, 2021
Editorial Decision:	June 8, 2021
Revision Received:	June 24, 2021
Accepted:	June 30, 2021

Editor: Cezar Khursigara

Reviewer(s): The following individuals involved in review of your submission have agreed to reveal their identity: Kathleen M. Scott (Reviewer #1)

Transaction Report:

DOI: <https://doi.org/10.1128/Spectrum.00305-21>

June 8, 2021

Prof. Friedrich Götz
University of Tübingen
Microbial Genetics, Interfaculty Institute of Microbiology and Infection Medicine (IMIT)
Auf der Morgenstelle 28
Tübingen 72076
Germany

Re: Spectrum00305-21 (The MpsAB bicarbonate transporter is superior to carbonic anhydrase in biofilm-forming bacteria with limited CO₂ diffusion)

Dear Prof. Friedrich Götz:

Thank you for submitting your manuscript to Microbiology Spectrum. As you will see the reviewers support publication of a revised paper. Please revise the paper along the lines suggested by the reviewers. When submitting the revised version of your paper, please provide (1) point-by-point responses to the issues raised by the reviewers as file type "Response to Reviewers," not in your cover letter, and (2) a PDF file that indicates the changes from the original submission (by highlighting or underlining the changes) as file type "Marked Up Manuscript - For Review Only". Please use this link to submit your revised manuscript - we strongly recommend that you submit your paper within the next 60 days or reach out to me. Detailed information on submitting your revised paper are below.

Link Not Available

Sincerely,

Cezar Khursigara

Journals Department
Reviewer comments:

Reviewer #1 (Comments for the Author):

This manuscript is a very interesting study of the interplay of carbonic anhydrase and dissolved inorganic carbon transporter presence in genomes of Firmicutes. The study does a nice job showing how CA or transporter presence can complement loss of one or the other in site directed mutants, and that cells that produce an elaborate extracellular matrix might be particularly beholden to dissolved inorganic carbon transporters to obtain sufficient dissolved inorganic carbon for growth. Specific suggestions on the manuscript follow:

Line 41: carbonic anhydrase doesn't really concentrate inorganic carbon, it merely facilitates its interconversion. Change 'carbon concentration systems' to 'systems for dissolved inorganic carbon supply'?

Line 57: replace 'as bicarbonate transporter has an advantage' with 'expressing bicarbonate transporters provides an advantage'

Line 64 and elsewhere (e.g., line 88): see comment above about line 41

Line 89 some carbonic anhydrase enzymes are found in the periplasm, and some are membrane-associated

Line 89 "with a mostly a Zn-binding domain" should be reworded as "most of which have a Zn-binding domain"

Line 91 CA activity, since it can also convert bicarb to CO₂, does not trap bicarb in the cytoplasm. One of the early experiments to 'prove' that the dissolved inorganic carbon pool in the cytoplasm of cyanobacteria consisted predominantly of bicarbonate and was out of chemical equilibrium consisted of expressing human CA in a cyanobacterium expressing bicarbonate transporters. When this was done, these mutant cells became "CO₂ fountains"---vast quantities of cytoplasmic dissolved inorganic carbon were lost as CO₂

Line 97 "In this process, HCO₃⁻ is continuously removed from the chemical equilibrium of the external milieu and at the same time it is continuously replenished." Not sure what is meant here?

Table S2 and elsewhere-eukaryotic and prokaryotic CA aren't the best descriptors, despite the Pfam names, as there are representatives from each group that have the 'wrong' CA. I'd keep the Pfam numbers (e.g., Pfam00484) but ditch the names for ones that more adequately describe the evolutionary history of these enzymes. 'Prokaryotic CA' enzymes are beta-class carbonic anhydrases; 'Eukaryotic CA' enzymes are alpha-class carbonic anhydrases.

Did you look for gamma CA?

Line 531 did you mean "The respective colonies harboring the right genes"? Not sure what is meant here

Line 588 Replace 'expect' with 'except'

Line 598 Replace 'avoid the cells clump formation due to possible biofilm' with 'to disrupt clumps'

Line 642 replace 'use' with 'used'

Supplemental figure 1 caption-A and B are mislabeled; I think the "A" in the figure is skipped in the caption. Also rephrase "CA is mostly point to the right" as "CA genes mostly point to the right"

Line 191 "No significant homology to any other proteins was found, implying that only a single CA is present in each strain." Since the different classes of carbonic anhydrase have independent evolutionary origins, using a BLASTp search with a beta CA would not get any alpha CA hits (or vice versa, or gamma CA hits, et cetera). Rephrase this.

Line 206 'is the ligation of the zinc active site with sulfur atoms...' could be rephrased as "facilitates the ligation of zinc in the active site, with sulfur atoms.."

Line 43, figure 2 caption needs a reference to fig 2B.

Line 291 'are correlated'

Supplementary fig 4-any ideas why deletion of mpsABC improved growth at pH 7.5 in *S. aureus*?

Line 377 why would aerobic organisms have a higher requirement for anaplerotic reactions than anaerobic ones? Is this because the aerobic ones would have a complete oxidative citric acid cycle, while the anaerobic (presumably fermentative ones) would not? If so, clarify for the reader

Line 384 there have been a number of good studies on CA in other members of Bacteria; not sure 'few' fits. Other species, just for alpha-CA:

Mesorhizobium loti

Rhodopseudomonas palustris

Helicobacter pylori

Thiomicrospira crunogena

And others. I think *Neisseria* might be in there too

Plus all the work on carboxysomal carbonic anhydrase

There are places in the discussion that could be shortened, that basically repeat results instead of interpreting them. Parts of the paragraph beginning on line 397 could be abbreviated to avoid repeating the results section and refocus the paragraph on interpretation. This is also true for the paragraph beginning on line 446. This could be accomplished by modifying the first few sentences of this paragraph. The same is true for the paragraph beginning on line 465

Line 459 "attributed to the evolution factor" needs to be rephrased. I would delete "can be attributed to the evolution factor, for example mpsAB can be acquired for group (iv) by horizontal gene transfer" and replace it with "could be attributed to acquisition of mpsAB by horizontal gene transfer"

Reviewer #2 (Comments for the Author):

The manuscript entitle "The MpsAB bicarbonate transporter is superior to carbonic anhydrase in

biofilm-forming bacteria with limited CO₂ diffusion" (ID: Spectrum00305-21) was reviewed carefully. The authors paid a detailed attention to both MpsAB and CA in *Staphylococcus aureus* and *S. carnosus* to explore their importance in CO₂ uptake and bicarbonate transfer. Despite of the great attractions of the subject and performed studies, big questions and doubts have raised as following that must be solved by the authors of this study:

1- Line 80: CO₂/bicarbonate (HCO₃⁻)?!!! It must be without CO₂.

2- Line 81: the correct is "eukaryotes".

3- Line 89: "CAs are a cytoplasmic enzyme". Generally, CAs can be localized in other subcellular locations like membrane, mitochondria, and nucleus as well as cytoplasm. In addition, CAs are classified to eight evolutionary families, not classes. In addition, it would be better to state what kind of CA families are present in prokaryotes. You can use these Refs for more information and citation: "PMID: 29802189 and PMID: 32393172".

4- Line 116: "Gammaproteobacteria" should be non-italic.

5- Line 116 and line 118: we have two different form of writing: "Gammaproteobacteria" and "γ-proteobacterial". The text must be uniform. One form the beginning to end.

6- Line 124: "Firmicutes" must be non-italic.

7- Line 128: "S. aureus possess only the MpsAB transporter, but no CA". How did you find there is no CA in *S. aureus*? This statement was mentioned again in lines 134-135. Again, there is this mistake in Table 3.

Based on my analyses, there is one beta CA in *S. aureus* as following:

>MVW54107.1 carbonic anhydrase, partial [*Staphylococcus aureus*]

```
LLAGNVRFGGKTSPKDYLVRSQQVAEQHPYAAVLACADSRLSPEILFDESLGKLFVVRTAGHVVDPA  
LGSIEYAVEHLHVNLLFVLGHESCGAVKATIGGGEAPPNIKALLRRIRPAVEKAHSQDLPEKDLLNACIK  
ENVRYQMQEAL
```

In addition, there are several gamma CAs in *S. aureus* as following (three sequences as the examples):

>NGG14433.1 gamma carbonic anhydrase family protein, partial [*Staphylococcus aureus*]

```
MSDTLRPYKNLFPGIGQRVMIDTSSVVIGDVRLADDVGIWPLVVIRGDVNYVAIGARTNIQDGSVLHVTH  
KSSSNPHGNPLIIGEDVTVGHKVILHG
```

>NGB42162.1 gamma carbonic anhydrase family protein [*Staphylococcus aureus*]

```
MSDTLRPYKNLFPGIGQRVMIDTSSVVIGDVRLADDVGIWPLVVIRGDVNYVAIGARTNIQDGSVLHVTH  
KSSSNPHGNPLIIGEDVTVGHKVMLHGCTIGNRVLVGMGSVLDGAIIEDDVMIGAGSLVPQHKRLESGY  
LYLGSPVKQIRPLSDAERSGLQYSANNYVKWKDDYLSQDNHIQP
```

>WP_094666538.1 gamma carbonic anhydrase family protein, partial [*Staphylococcus aureus*]

```
MERFIASNATVIGDVTLSEDVTWYQAVLRGDSNWIKIGQRTNIQDGTIIHVDHDAPVDIAENVTVGHC  
MLHGCTIEKGALIGMGTIILNHAVIGENSLIGAGSLVTEGKVIPPVLAFFGRPAKVIRPLTKEEQKKE  
NIQHYVEIG
```

Even, it was mentioned there is no CA in *S. aureus* in a "Nature Communication" paper that is not correct: Table 1 in "PMID: 31399577", which the first author and the corresponding author are similar to this manuscript. A big mistake was published in the "Nature Communication" in 2019 and another one has submitted to a journal from ASM now.

8- Line 136: the CA from *Staphylococcus carnosus* belongs to what CA family that was mentioned in Figure 1 as well? Why? Please indicate in the manuscript.

Staff Comments:

Preparing Revision Guidelines

For complete guidelines on revision requirements, please see the Instructions to Authors at [link to page]. **Submissions of a paper that does not conform to Microbiology Spectrum guidelines will delay acceptance of your manuscript.**

Please return the manuscript within 60 days; if you cannot complete the modification within this time period, please contact me. If you do not wish to modify the manuscript and prefer to submit it to another journal, please notify me of your decision immediately so that the manuscript may be formally withdrawn from consideration by Microbiology Spectrum.

If you would like to submit an image for consideration as the Featured Image for an issue, please contact Spectrum staff.

Dear Editor,

The manuscript entitle "The MpsAB bicarbonate transporter is superior to carbonic anhydrase in biofilm-forming bacteria with limited CO₂ diffusion" (ID: Spectrum00305-21) was reviewed carefully. The authors paid a detailed attention to both MpsAB and CA in *Staphylococcus aureus* and *S. carnosus* to explore their importance in CO₂ uptake and bicarbonate transfer. Despite of the great attractions of the subject and performed studies, big questions and doubts have raised as following that must be solved by the authors of this study:

- 1- Line 80: CO₂/bicarbonate (HCO₃⁻)?!!! It must be without CO₂.
- 2- Line 81: the correct is "eukaryotes".
- 3- Line 89: "CAs are a cytoplasmic enzyme". Generally, CAs can be localized in other subcellular locations like membrane, mitochondria, and nucleus as well as cytoplasm. In addition, CAs are classified to eight evolutionary families, not classes. In addition, it would be better to state what kind of CA families are present in prokaryotes. You can use these Refs for more information and citation: "PMID: 29802189 and PMID: 32393172".
- 4- Line 116: "Gammaproteobacteria" should be non-italic.
- 5- Line 116 and line 118: we have two different form of writing: "Gammaproteobacteria" and "γ-proteobacterial". The text must be uniform. One form the beginning to end.
- 6- Line 124: "Firmicutes" must be non-italic.
- 7- Line 128: "*S. aureus* possess only the MpsAB transporter, but no CA". How did you find there is no CA in *S. aureus*? This statement was mentioned again in lines 134-135. Again, there is this mistake in Table 3.

Based on my analyses, there is one beta CA in *S. aureus* as following:

```
>MVW54107.1 carbonic anhydrase, partial [Staphylococcus aureus]
LLAGNVRVFGGKTS PKDYLVERSQQVAEQHPYAAVLACADSRLSPEILFDESLGKLFVVRTAGHVVDPA
LGSIEYAVEHLHVNLLFVLGHESCGAVKATIGGGEAPPNIKALLRRIRPAVEKAHSQDLPEKDLLNACIK
ENVRYQMQEAL
```

In addition, there are several gamma CAs in *S. aureus* as following (three sequences as the examples):

```
>NGG14433.1 gamma carbonic anhydrase family protein, partial
[Staphylococcus aureus]
MSDTLRPFYKNLFPGIGQRMIDTSSVVIGDVRLADDVGIWPLVVIRGDVNYVAIGARTNIQDGSVLHVTH
KSSSNPHGNPLIIGEDVTVGHKVLHG
```

```
>NGB42162.1 gamma carbonic anhydrase family protein [Staphylococcus aureus]
MSDTLRPFYKNLFPGIGQRMIDTSSVVIGDVRLADDVGIWPLVVIRGDVNYVAIGARTNIQDGSVLHVTH
KSSSNPHGNPLIIGEDVTVGHKVMLHGCTIGNRVLVGMGSIVLDGAIIEDDVMIGAGSLVPQHKRLESGY
LYLGSVPVKQIRPLSDAERSGLQYSANNYVKWKDDYLSQDNHIQP
```

```
>WP_094666538.1 gamma carbonic anhydrase family protein, partial
[Staphylococcus aureus]
MERFIASNATVIGDVTLSQVDTIWIYQAVLRGDSNWIKIGQRTNIQDGTIIHVDHDAPVDIAENVTVGHQC
MLHGCTIEKGALIGMGTIILNHAVIGENSLIGAGSLVTEGKVIIPPVNLAFGRPAKVIRPLTKKEIQKNKE
NIQHYVEIG
```

Even, it was mentioned there is no CA in *S. aureus* in a "Nature Communication" paper that is not correct: Table 1 in "PMID: 31399577", which the first author and the corresponding

author are similar to this manuscript. A big mistake was published in the "Nature Communication" in 2019 and another one has submitted to a journal from ASM now.

- 8- Line 136: the CA from *Staphylococcus carnosus* belongs to what CA family that was mentioned in Figure 1 as well? Why? Please indicate in the manuscript.

In the current form, this study is not recommended for publication at "Microbiology Spectrum" and even transfer to other sister journals from ASM. The general subject for this manuscript is attractive and can be re-submitted after extensive modification on it.

Best wishes

Reviewer comments:

Reviewer #1 (Comments for the Author):

This manuscript is a very interesting study of the interplay of carbonic anhydrase and dissolved inorganic carbon transporter presence in genomes of Firmicutes. The study does a nice job showing how CA or transporter presence can complement loss of one or the other in site directed mutants, and that cells that produce an elaborate extracellular matrix might be particularly beholden to dissolved inorganic carbon transporters to obtain sufficient dissolved inorganic carbon for growth. Specific suggestions on the manuscript follow:

Line 41: carbonic anhydrase doesn't really concentrate inorganic carbon, it merely facilitates its interconversion. Change 'carbon concentration systems' to 'systems for dissolved inorganic carbon supply'?

We really thank you for this suggestion. We also think that 'carbon concentrating system' is not really correct. We changed all according to your suggestion: 'systems for 'dissolved inorganic carbon supply' (DICS). Please see line 40 and other changes highlighted in yellow.

Line 57: replace 'as bicarbonate transporter has an advantage' with 'expressing bicarbonate transporters provides an advantage'

Thank you for the suggestion. We have amended it accordingly. Please see line 56.

Line 64 and elsewhere (e.g., line 88): see comment above about line 41

We have replaced 'carbon concentration systems' to 'systems for dissolved inorganic carbon supply' throughout the manuscript. Please see the changes highlighted in yellow.

Line 89 some carbonic anhydrase enzymes are found in the periplasm, and some are membrane-associated

Thank you for pointing out the mistake. We have corrected the sentence to "CAs are ubiquitous enzymes which can be found in the mitochondria, cytoplasm, periplasm, membrane or cell wall-associated, carboxysome and also chloroplast in plants (3, 4)". Please see line 89.

Line 89 "with a mostly a Zn-binding domain" should be reworded as "most of which have a Zn-binding domain"

Thank you for the comment. We have corrected the sentence to "Most of these enzymes have a Zn-binding domain that..... Please see line 91.

Line 91 CA activity, since it can also convert bicarb to CO₂, does not trap bicarb in the cytoplasm. One of the early experiments to 'prove' that the dissolved inorganic carbon pool in the cytoplasm of cyanobacteria consisted predominantly of bicarbonate and was out of chemical equilibrium consisted of expressing human CA

in a cyanobacterium expressing bicarbonate transporters. When this was done, these mutant cells became "CO₂ fountains"---vast quantities of cytoplasmic dissolved inorganic carbon were lost as CO₂

Thank you for the comment. We removed the 'trapping part' and rephrase the sentence: CA is important for inorganic carbon fixing-enzymes which utilizes either CO₂ or HCO₃⁻ by interconverting these species to ensure sufficient concentration in the cytoplasm. Please see line 93.

Line 97 "In this process, HCO₃⁻ is continuously removed from the chemical equilibrium of the external milieu and at the same time it is continuously replenished." Not sure what is meant here?

We apologized for the confusion. We have rephrased the sentence: Such a transporter transports HCO₃⁻ from the external environment over the membrane into the cytoplasm, where the imported HCO₃⁻ is consumed by the carboxylation reactions. The continuous consumption of HCO₃⁻ in the cytoplasm could induce a suction power to keep the transporter running. In the exterior milieu, the transporter is continuously removing HCO₃⁻ from the CO₂ / HCO₃⁻ equilibrium resulting in a permanent replenishment of HCO₃⁻. Please see line 103.

Table S2 and elsewhere-eukaryotic and prokaryotic CA aren't the best descriptors, despite the Pfam names, as there are representatives from each group that have the 'wrong' CA. I'd keep the Pfam numbers (e.g., Pfam00484) but ditch the names for ones that more adequately describe the evolutionary history of these enzymes. 'Prokaryotic CA' enzymes are beta-class carbonic anhydrases; 'Eukaryotic CA' enzymes are alpha-class carbonic anhydrases.

Thank you for the comment. We agreed, too, the eukaryotic and prokaryotic CA are not the best descriptor but we used the Pfam names just because they are named as such. However, we would follow your suggestion to keep the Pfam numbers for the Tables and put 'Prokaryotic CA' as β-CA instead and 'Eukaryotic CA' as α-CA. Please see the changes highlighted in yellow in table 1, Figure S1, table S1 and the manuscript and we included a sentence in line 166 regarding this.

Did you look for gamma CA?

Yes. We blasted the α and γ-CAs listed below against *S. aureus*, *S. carnosus* and *S. pseudintermedius* and we did not find any homology. Below are the BLASTp results. As we have reached the limit of supplementary materials (max 10), we did not include it there. However, now we include the info in the manuscript, "For *S. carnosus* and *S. pseudintermedius* genomes, we did not find any other homology (BLASTp) with α-CAs from *Enterococcus faecium*, *Helicobacter pylori*, *Neisseria gonorrhoea*, *Vibrio cholerae* and human CA1 and CA2 as well as γ-CAs from *Enterococcus faecium*, *Escherichia coli*, *Halobacterium salinarum* and *Methanosarcina thermophila*." Please see line 206. We also mentioned this for *S. aureus*: "In addition, we also did not find significant homology (BLASTp) with α-CAs from *Enterococcus faecium*, *Helicobacter pylori*, *Neisseria gonorrhoea*, *Vibrio cholerae* and human CA1 and CA2 as well as γ-CAs from *Enterococcus faecium*,

Escherichia coli, *Halobacterium salinarum* and *Methanosarcina thermophila*.”
Please see line 377:

As can be seen from our rebuttal to reviewer #2, *S. aureus* does not have any CA genes/proteins. The publications that postulate the presence of CA in *S. aureus* are highly questionable, as shown in the response to reviewer 2.

Sequence homology search for selected α -CAs in the genomes of *S. aureus*, *S. carnosus* and *S. pseudintermedius* using BLASTp

Species	Accession code (Uniprot)	Length (of a.a)	S. aureus (NCBI taxid:1280)	S. carnosus (NCBI taxid: 1281)	S. pseudintermedius (NCBI taxid: 283734)
Enterococcus faecium	Q3XYE8	234	No significant similarity found		
Helicobacter pylori	A0A0M3KL20	234	No significant similarity found		53/54 identity (98%) (WP_181892146.1) [§] in only one unfinished genome of S. pseudintermedius strain ST525
Neisseria gonorrhoea	Q50940	252	No significant similarity found		
Vibrio cholerae	Q9KMP6	239	No significant similarity found		18/49 identity (37%) (WP_181892146.1) ^{§§} in only one unfinished genome of S. pseudintermedius strain ST525
Human CA1	P00915	261	27/72 identity (38%) (MBO8666615.1) in one unfinished genome of S. aureus strain IHMA56* and another hit 22/60 identity (37%) (MBO8619751.1) in one unfinished genome of S. aureus strain IHMA68**	No significant similarity found	
Human CA2	P00918	260	27/54 identity (50%) (MBO8619751.1) in one unfinished genome of S. aureus strain IHMA68 ^v and another hit 24/75	No significant similarity found	

			identity (37%) (MBO8666615.1) in one unfinished genome of S. aureus strain IHMA56 ^{ψψ}	
--	--	--	---	--

ξ However, when this protein **WP_181892146.1** is blasted in NCBI BLASTp, it has only one hit as *S. aureus* (its own sequence) and the rest of the 99 hits are from *H. pylori* with 100% identity.

ξξ The same protein **WP_181892146.1** was found as above.

* However, when this protein **MBO8666615.1** is blasted in NCBI BLASTp, it has only one hit as *S. aureus* (its own sequence), along with 99 hits of 100% in various organisms, including 100% identity in *Homo sapiens*. Therefore this sample is most likely contaminated with human specimen.

** When this protein **MBO8619751.1** is blasted in NCBI BLASTp, it has only one hit as *S. aureus* (its own sequence) and 99 other hits from various organisms with 81-91% identity. Therefore, this sample is most likely contaminated with sources from other organisms.

ψ When this protein **MBO8619751.1** is blasted in NCBI BLASTp, it has only one hit as *S. aureus* (its own sequence) and 99 other hits from various organisms with 81-91% identity. Therefore, this sample is most likely contaminated with sources from other organisms.

ψψ When this protein **MBO8666615.1** is blasted in NCBI BLASTp, it has only one hit as *S. aureus* (its own sequence) and 99 other hits of 100% in various organisms, including 100% identity in *Homo sapiens*. Therefore this sample is most likely contaminated with human specimen.

Sequence homology search for selected γ -CAs in the genomes of *S. aureus*, *S. carnosus* and *S. pseudintermedius* using BLASTp

Species	Accession code (Uniprot)	Length (of a.a)	S. aureus (NCBI taxid:1280)	S. carnosus (NCBI taxid: 1281)	S. pseudintermedius (NCBI taxid: 283734)
Enterococcus faecium	Q3XX77	161	3 hits in S. aureus as γ -CAs but these protein sequences (NGG14433.1 , NGB42162.1 and WP_094666538.1) are most likely contaminated with	No significant similarity found	

			other organism*, the rest of the hits are from S. aureus proteins annotated as phenylacetic acid degradation protein PaaY (32/83 identity) and/or sugar O-acetyltransferase. Please see the footnote below.*	
Escherichia coli	P0A9W9	184	3 hits in S. aureus as γ-CAs but these protein sequences are most likely contaminated with other organism* (same proteins NGG14433.1, NGB42162.1 and WP_094666538.1 as in above) and another as phenylacetic acid degradation protein PaaY in S. aureus.	No significant similarity found
Methanosarcina thermophila	P40881	247	3 hits in S. aureus as γ-CAs but these protein sequences are most likely contaminated with other organism* (same proteins NGG14433.1, NGB42162.1 and WP_094666538.1 as in above).	No significant similarity found
Halobacterium salinarum	Q9HR64	220	51/163 identity (31%) (MVW54107.1) in only one unfinished genome of S. aureus strain mecC 165 PE. Please see the footnote below.*	No significant similarity found

*** Please refer to the explanation for these 4 proteins sequences for reviewer #2.**

Line 531 did you mean "The respective colonies harboring the right genes"? Not sure what is meant here

We apologized for the error. We have corrected it to: "The respective clones harboring the right genes.....Please see line 535.

Line 588 Replace 'expect' with 'except'

We apologized for the typo. Please see the correction at line 591.

Line 598 Replace 'avoid the cells clump formation due to possible biofilm' with 'to disrupt clumps'

Thank you for the suggestion. Please see line 601.

Line 642 replace 'use' with 'used'

We apologized for the mistake. Please see line 645.

Supplemental figure 1 caption-A and B are mislabeled; I think the "A" in the figure is skipped in the caption. Also rephrase "CA is mostly point to the right" as "CA genes mostly point to the right"

Thank you for highlighting the errors. We have amended it accordingly. Please see line 33 and 43.

Line 191 "No significant homology to any other proteins was found, implying that only a single CA is present in each strain." Since the different classes of carbonic anhydrase have independent evolutionary origins, using a BLASTp search with a beta CA would not get any alpha CA hits (or vice versa, or gamma CA hits, et cetera). Rephrase this.

Thank you for the comment. We have included 'β-CA' since the CAs from *S. carnosus* and *S. pseudintermedius* cloned in *E. coli*Δ*can* are from β-CA class: "No significant homology to any other proteins was found, implying that only a single β-CA is present in each strain." Please see line 202. We have also included a sentence that BLASTp showed no homology with α- and γ-CAs from other bacteria. Please refer to the above reply and line 206.

Line 206 'is the ligation of the zinc active site with sulfur atoms...' could be rephrased as "facilitates the ligation of zinc in the active site, with sulfur atoms.."

Thank you for the correction. Please see line 221.

Line 43, figure 2 caption needs a reference to fig 2B.

Thank you for the carefully checking the manuscript. Please see line 907 for the correction.

Line 291 'are correlated'

Please see line 305.

Supplementary fig 4-any ideas why deletion of mpsABC improved growth at pH 7.5 in *S. aureus*?

We apologized if the figure is not clear. We mentioned it in the figure legend that lower OD in figure 4C was due to clumps/cell agglutinations. The readings were recorded automatically by a microplate reader. We have now included an arrow with the word 'clumps' in Figure 4C. Please refer to Supplementary Figure 4C and line 130 in the figure legend.

Line 377 why would aerobic organisms have a higher requirement for anaplerotic reactions than anaerobic ones? Is this because the aerobic ones would have a complete oxidative citric acid cycle, while the anaerobic (presumably fermentative ones) would not? If so, clarify for the reader

Thank you for the intriguing questions. It gave us another perspective to the interpretation, which probably might be more plausible. we deleted this statement from the manuscript and added another interpretation in the discussion: "In principle, it does not really sense if the bacteria have both a HCO_3^- transporter and a CA because the transported HCO_3^- will be converted by CA to CO_2 which can escape the cells by diffusion. Thus, the benefit of a transporter is mitigated by the presence of a CA. This is exactly what we observed experimentally by co-expressing *can* in *S. aureus* (pRB473-*canSc*) which caused a decreased growth compared to the wild type. Indeed, most of the bacterial species have only one or the other system. Nevertheless, there are a few species which have both systems like some endospore-forming bacilli and clostridia (**Table 1**). In such cases where microorganisms undergo morphological differentiation, it might be advantageous to have both systems, where both systems are active in different cell compartments." Please see line 424.

Line 384 there have been a number of good studies on CA in other members of Bacteria; not sure 'few' fits. Other species, just for alpha-CA:

Mesorhizobium loti

Rhodopseudomonas palustris

Helicobacter pylori

Thiomicrospira crunogena

And others. I think *Neisseria* might be in there too

Plus all the work on carboxysomal carbonic anhydrase

Thank you for the comment. We have included β -CA in that sentence and added some studies of α and γ -CAs in other bacteria. "On another note, there are only some bacteria in which the β -CA activity was thoroughly described to date (3) such as in *E. coli* (19), *Streptococcus pneumoniae* (18) or *Clostridium autoethanogenum* (27). In other classes, the α -CA includes studies in *Helicobacter pylori* (28), *Neisseria gonorrhoeae* (29), *Mesorhizobium loti* (30) and *Rhodopseudomonas palustris* (31) while γ -CA was well-characterized in *Methanosarcina thermophila* (32). Different classes of CAs (α and β) have been reported in the same bacteria such as *Thiomicrospira crunogena* (33)". Please see line 402.

There are places in the discussion that could be shortened, that basically repeat results instead of interpreting them. Parts of the paragraph beginning on line 397 could be abbreviated to avoid repeating the results section and refocus the paragraph on interpretation. This is also true for the paragraph beginning on line 446. This could be accomplished by modifying the first few sentences of this paragraph. The same is true for the paragraph beginning on line 465

We have shortened the Discussion as suggestion. Please see changes highlighted in yellow from line 417-434, 459-469 and 471-474.

Line 459 "attributed to the evolution factor" needs to be rephrased. I would delete "can be attributed to the evolution factor, for example mpsAB can be acquired for group (iv) by horizontal gee transfer" and replace it with "could be attributed to acquisition of mpsAB by horizontal gene transfer"

Thank you for the suggestion. We have amended it as suggested: The lack of a strict correlation between the last two groups could be attributed to acquisition of *mpsAB* or *ica* by horizontal gene transfer. Please see line 464.

Reviewer #2 (Comments for the Author):

The manuscript entitle "The MpsAB bicarbonate transporter is superior to carbonic anhydrase in biofilm-forming bacteria with limited CO₂ diffusion" (ID: Spectrum00305-21) was reviewed carefully. The authors paid a detailed attention to both MpsAB and CA in *Staphylococcus aureus* and *S. carnosus* to explore their importance in CO₂ uptake and bicarbonate transfer. Despite of the great attractions of the subject and performed studies, big questions and doubts have raised as following that must be solved by the authors of this study:

1- Line 80: CO₂/bicarbonate (HCO₃⁻)?!!! It must be without CO₂.

We apologized for the confusion. We have corrected the sentence to be "CO₂ and bicarbonate (HCO₃⁻)". Please see line 38, 61 and 80.

2- Line 81: the correct is "eukaryotes".

We have corrected the typo. Please see line 81.

3- Line 89: "CAs are a cytoplasmic enzyme". Generally, CAs can be localized in other subcellular locations like membrane, mitochondria, and nucleus as well as cytoplasm.

Thank you for pointing out the mistake. We have corrected the sentence to "CAs are ubiquitous enzymes which can be found in the mitochondria, cytoplasm, periplasm, membrane or cell wall-associated, carboxysome and also chloroplast in plants (3, 4)". Please see line 89.

In addition, CAs are classified to eight evolutionary families, not classes. In addition, it would be better to state what kind of CA families are present in prokaryotes. You can use these Refs for more information and citation: "PMID: 29802189 and PMID: 32393172".

Thank you for the suggestion. CA classes and families are used interchangeably in most of the publications. In fact, one of the most cited publication on CA, "Prokaryotic carbonic anhydrases" by Smith and Ferry (2000) used the term "class" to describe the different groups of CA. We have added the sentence "To date, CAs from eight evolutionary distinct families have been reported (α , β , γ , δ , ζ , η , θ , and ι) (8-11)". Please see line 95-98.

4- Line 116: "Gammaproteobacteria" should be non-italic.

We have corrected the mistake. Please see line 123.

5- Line 116 and line 118: we have two different form of writing: "Gammaproteobacteria" and " γ -proteobacterial". The text must be uniform. One form the beginning to end.

Thank you for the comment. We have standardized the term "gamma" as γ in the text. Please see line 123 and 125.

6- Line 124: "Firmicutes" must be non-italic.

Thank you for the comment. We have amended it accordingly throughout the manuscript. Please see the changes highlighted in yellow.

7- Line 128: "S. aureus possess only the MpsAB transporter, but no CA". How did you find there is no CA in S. aureus? This statement was mentioned again in lines 134-135. Again, there is this mistake in Table 3.

As mentioned in the manuscript (line 367-381), we concluded that there is no CA in *S. aureus* based on our extensive search using 209 finished genomes sequences in IMG/G database. Since we confirmed the presence of an functional CA in *S. carnosus* in this work, we blasted this sequence against all 209 genomes sequences but did not find any similarity. We also searched this database for any protein annotated as CA or putative CA and also searched on the basis of two Pfam motifs related to CA but no similarity was found.

In addition, we also blasted the *S. carnosus* CA against *S. aureus* in NCBI database and found 2 hits, Sequence ID: SPZ78436.1 (193 amino acids) and Sequence ID: SPZ78435.1 (61 amino acids) annotated as CA from *S. aureus*. SPZ78436.1 is from a *S. aureus* strain NCTC12981 which is not fully assembled yet. A BLAST of this protein on NCBI only showed ONE 100% hit with its own sequence (*S. aureus*) while there are multiple hits from other Staphylococcus species. The highest similarity is from *S. schleiferi* which covers 69% of the length with 100% identity (e value 5e-94). The second protein, SPZ78435.1 also originates from the same strain, NCTC12981. A BLASTp of this protein also revealed that it has ONE 100% hit with its own sequence (*S. aureus*) while the multiple hits from other Staphylococcus species, the highest similarity from *S. coagulans* (86% coverage with 98% identity, e value 1e-27). Clearly, this shows that the protein sequences from unassembled sequences are not accurate and unreliable and therefore we ignored it. For this reason, we restricted our bioinformatics analyses to only fully assembled genomes. We did not simply take a protein sequence from NCBI database but from finished genomes and validated reference genome like *S. aureus* NCTC 8325 instead. We also included multiple sequence alignment with experimentally proven CAs (Figure 1), gene synteny comparison (Figure S1) and also phylogenetic analysis (Figure S7) to support our findings. Furthermore, we have also blasted the genome of *S. aureus* against the α and γ -CAs from bacteria in which the CAs activity was experimentally proven. Please refer to the reply for reviewer #1 for the BLASTp results.

Based on my analyses, there is one beta CA in *S. aureus* as following:

```
>MVW54107.1 carbonic anhydrase, partial [Staphylococcus aureus]
LLAGNVRFVGGKTSPKDYLVERSQQVAEQHPYAAVLACADSRLSPEILFDESLGKL
FVVRTAGHVVDPA
LGSIEYAVEHLHVNLLFVLGHESCGAVKATIGGGEAPPNIKALLRRIRPAVEKAHSQ
DLPEKDLLNACIK
ENVRYQMQEAL
```

In addition, there are several gamma CAs in *S. aureus* as following (three sequences as the examples):

>NGG14433.1 gamma carbonic anhydrase family protein, partial [Staphylococcus aureus]
MSDTLRPYKNLFPGIGQRVMIDTSSVIGDVRLADDVGIWPLVVIRGDVNYVAIGAR
TNIQDGSVLHVTH
KSSSNPHGNPLIIGEDVTVGHKVILHG

>NGB42162.1 gamma carbonic anhydrase family protein [Staphylococcus aureus]
MSDTLRPYKNLFPGIGQRVMIDTSSVIGDVRLADDVGIWPLVVIRGDVNYVAIGAR
TNIQDGSVLHVTH
KSSSNPHGNPLIIGEDVTVGHKVMLHGCTIGNRVLVGMGSIVLDGAIIEDDVMIGAG
SLVPQHKRLESGY
LYLGSPVKQIRPLSDAERSGLQYSANNYVKWKDDYLSQDNHIQP

>WP_094666538.1 gamma carbonic anhydrase family protein, partial
[Staphylococcus aureus]
MERFIASNATVIGDVTLSERVEDVTIWYQAVLRGDSNWIKIGQRTNIQDGTIIHVDHDAPV
DIAENVTVGHC
MLHGCTIEKGALIGMGTTLNHAVIGENSLIGAGSLVTEGKVIPPVLAFAFRPAKVIRP
LTKEEIQKNKE
NIQHYVEIG

To prove our point mentioned above, we used the 4 CAs provided by reviewer #2 and performed an extensive BLASTp. First, we blasted all 4 protein sequences from all 259 finished genomes of *S. aureus* in the IMG/G database which was updated recently. We used this database instead of NCBI because it is more organized to perform search and we could select the exact strains from either finished genomes, permanent draft or drafts. This enabled us to get more 'hits' compared to NCBI which is limited the first 100 sequences producing significant alignments. All 259 genomes showed no identity at all (no hits) for either **MVW54107.1**, **NGG14433.1** and **NGB42162.1**. The finding that there is no similarity in any of the strains already raised doubts about the authenticity of the protein sequences.

With **WP_094666538.1**, all 259 strains showed low identity, which was 33% with 40/122 amino acids aligned. These proteins were annotated as either acetyltransferase (isoleucine patch superfamily), acetyltransferase-like (isoleucine patch superfamily), galactoside O-acetyltransferase or hypothetical protein because they have the related COG, KOG or Pfam motifs. If **WP_094666538.1** is indeed from *S. aureus* or even if it is wrongly annotated, it would not be found in such a low identity in all 259 finished genomes, including some reference genomes. For example, one of the protein of our bicarbonate transporter operon, MpsB which is annotated as uncharacterized protein or YbcC family protein (WP_000211540.1, 901 amino acids) has 100 BLAST hits (maximum hits) with 99.78-99.89 % identity with all *S. aureus* strains covering 100% of the protein (901/901 amino acids).

To further confirm our findings, we performed the same search (BLAST) with **4590 permanent draft sequences of *S. aureus***.

Protein	Identity/hits (amino acids aligned)	Comments
MVW54107.1 (151 a.a)	No identity in all 4586 strains except 4 strains: 1. DEU37: 30% identity (49/163) as CA, partial gene with “no stop” 2. DEU28: 30% identity (49/163) as CA 3. DEU35: 33% identity (39/117) as CA, partial gene, “no start” 4. DEU41: 33% identity (39/117) as CA, partial gene, “no start”	All the 4 strains listed here were tagged as “ anomalous assembly: contaminated ” by NCBI
NGG14433.1 (97 a.a)	No identity in all 4583 strains except: 1. DEU28 2. DEU35 3. DEU37 or 4. DEU41 5. DEU42 6. DEU39 35% identity (27/77) as CA acetyltransferase partial gene, “no start” 7. C0673: 41% identity (38/92) as CA or acetyltransferase encoded by gene with locus tag V070_00826	All the 6 strains listed here were tagged as “ anomalous assembly: contaminated ” by NCBI *C0673 is wrongly annotated as S. aureus in NCBI database
NGB42162.1 (184 a.a)	No identity in 4584 strains except: 1. DEU28 2. DEU35 or 3. DEU37 4. DEU41 5. DEU42 5. DEU39 37% identity (57/156) as CA acetyltransferase 36% (31/86) as transferase hexapeptide (six repeat-containing protein) 6. C0673: 39% identity (67/171) as CA or acetyltransferase encoded by gene with locus tag V070_00826	All the 6 strains listed here were tagged as “ anomalous assembly: contaminated ” by NCBI *C0673 is wrongly annotated as S. aureus in NCBI database
WP_094666538.1 (149 a.a)	All 4590 strains showed similar results as with the finished genomes; that is 33%	These strains have the same similarity as all the

	identity (40/122) as either acetyltransferase (isoleucine patch superfamily), galactoside O-transferase or hypothetical protein. A few strains have unspecific hits, for example:  1. NRS384: 35% identity (33/95) as hexapeptide repeat of succinyltransferase 2. OCMM6067: 32% identity as (23/66) as 2,3,4,5 tetrahydropyridine-2-6, dicarboxylate N-acetyltransferase 3. 65-1322: 33% identity as (40/122) as transferase hexapeptide repeat containing protein 4. ATCC BAA-39: 33% identity (40/122) as galactoside-6-phosphate isomerase LacA subunit 5. C0673: 50% identity (72/144) as CA or acetyltransferase encoded by gene with locus tag V070_00826, 35% identity (26/77) as maltose O-acetyltransferase in gene with locus tag V070_00366 and 25% identity (26/103) as acetyltransferase (isoleucine patch superfamily) encoded by gene with locus tag V070_00906. 	finished genomes, as detailed in the text above. As there only a few strains out of 4590 permanent draft sequences with unspecific and a very low identity, we did not check the origins of each of these strains. If WP_094666538.1 is indeed from S. aureus, it should not be present in such a low identity in all 4819 S. aureus strains (259 finished and 4560 permanent draft sequences). There is not a single strain found with reasonable protein identity. *C0673 is wrongly annotated as S. aureus in NCBI database
--	---	--

* strain **C0673** showed multiple hits for **NGG14433.1**, **NGB42162.1** and **WP_094666538.1** gamma CAs in the same gene (locus tag V070_00826) and also V070_00366 and V070_00906. According to NCBI, this strain is an unfinished genome with 89 contigs where the taxonomy check is inconclusive. Although this strain is annotated as *S. aureus* C0673 in NCBI database, it is highly suspicious, so we decided to verify its genome sequence. We downloaded the sequence from NCBI (https://www.ncbi.nlm.nih.gov/genome/154?genome_assembly_id=158092) and checked it against Public databases for molecular typing and microbial genome diversity (PubMLST) https://pubmlst.org/bigsubdb?db=pubmlst_rmlst_seqdef_kiosk. According to PubMLST, **the predicted taxa for C0673 is actually *S. sciuri* (83%), now known as *Mammaliicoccus sciuri***. Using IMG/G website, we performed Pairwise Average Nucleotide Identity (ANI) check with 2 of the finished *S. sciuri* genomes. **C0673 has 97% nucleotide identity with *S. sciuri* SNUDS-18 and 96% nucleotide identity with *S. sciuri* FDAARGOS_285.** This strain is wrongly

annotated in NCBI database, which gave us “false positive” hits in our BLAST because *S. sciuri*, but not *S. aureus* has a CA according our analysis (Table 1 & Table 3).

Given the fact that not a single strain out of 4819 strains has a reasonable protein identity with any of the CA protein sequences provided by reviewer #2, we proceeded to examine the authenticity of these sequences.

1. >MVW54107.1 carbonic anhydrase, partial [*Staphylococcus aureus*]

This protein sequence comes from an unpublished, direct submission to NCBI from *Staphylococcus aureus* strain mecC 165 PE unfinished genome with 37 contigs. According to PubMLST, the predicted taxa is 100% *S. aureus*. However, when we blasted this sequence in NCBI, we only got one 100% hit which is against its own sequence. The rest of the 99 hits are from different bacteria like *Acidobacteria bacterium*, *Ignavibacteriales bacterium* etc with 47-91% identity as shown by the screenshot below, which **means that this supposedly CA is not even from *S. aureus***. This shows that the unfinished genomes are unreliable.

Description	Scientific Name	Max Score	Total Score	Query Cover	E value	Per. Ident	Acc. Len	Accession
carbonic anhydrase [Staphylococcus aureus]	Staphylococcus aur...	307	307	100%	2e-105	100.00%	151	MVW54107.1
hypothetical protein [Acidobacteria bacterium]	Acidobacteria bacte...	285	285	100%	1e-92	91.39%	443	MBI1762878.1
carbonic anhydrase [Ignavibacteriales bacterium]	Ignavibacteriales ba...	191	191	100%	5e-58	60.26%	271	MBI3578216.1
partial Carbonic anhydrase 2 [Anaerolineae bacterium]	Anaerolineae bacter...	191	191	98%	1e-57	61.74%	303	CAG0959001.1
hypothetical protein A2X60_18185 [Ignavibacteria bacterium GWF2_35_20]	Ignavibacteria bacte...	186	186	100%	1e-56	57.62%	225	OGU62690.1
carbonic anhydrase [Bacteroidetes bacterium]	Bacteroidetes bacte...	186	186	98%	2e-56	60.14%	222	MBP6671919.1
carbonic anhydrase [Acidobacteria bacterium]	Acidobacteria bacte...	188	188	98%	6e-55	59.46%	439	MBS1808876.1
carbonic anhydrase [Bacteroidetes bacterium]	Bacteroidetes bacte...	181	181	100%	5e-54	56.29%	279	MBP6671508.1
carbonic anhydrase [Bacteroidetes bacterium]	Bacteroidetes bacte...	175	175	100%	2e-51	54.97%	278	NUN68686.1
carbonic anhydrase [Ignavibacteriales bacterium]	Ignavibacteriales ba...	167	167	100%	1e-48	54.97%	251	MBD3411237.1
carbonic anhydrase [bacterium]	bacterium	162	162	100%	3e-47	53.29%	220	TAL17176.1
carbonic anhydrase [Ignavibacteriales bacterium]	Ignavibacteriales ba...	160	160	98%	1e-46	50.68%	216	RJP63275.1
TPA: carbonic anhydrase [Caldimicrobium sp.]	Caldimicrobium sp.	158	158	100%	9e-46	50.98%	203	HEN51088.1
carbonic anhydrase [Ignavibacteria bacterium]	Ignavibacteria bacte...	157	157	98%	2e-45	54.36%	214	MBN8544675.1
carbonic anhydrase [Clostridium formicaceticum]	Clostridium formicac...	158	158	98%	2e-45	49.67%	245	WP_070972436.1
carbonic anhydrase [Ignavibacteriales bacterium]	Ignavibacteriales ba...	155	155	98%	5e-45	52.35%	186	MBK7866995.1
carbonic anhydrase [bacterium]	bacterium	157	157	98%	8e-45	52.67%	243	

2. >NGG14433.1 gamma carbonic anhydrase family protein, partial [*Staphylococcus aureus*]

This protein sequence comes from an unpublished, direct submission to NCBI from *Staphylococcus aureus* strain UG271, unfinished genome with 397 contigs. According to PubMLST, the predicted taxa is 100% *S. aureus*. Again, when we blasted this sequence in NCBI, we only got one 100% hit which is against its own sequence. The rest of the 99 hits are all from *Salmonella enterica* with 99% identity

and 100% protein coverage. Again, this proves that **this so-called “gamma CA” is not even a protein from *S. aureus* but instead 100% from *Salmonella enterica*.**

Descriptions		Graphic Summary	Alignments	Taxonomy				
Sequences producing significant alignments								
Download		New Select columns		Show 100				
[x] select all 100 sequences selected GenPept Graphics Distance tree of results Multiple alignment New MSA Viewer								
Description	Scientific Name	Max Score	Total Score	Query Cover	E value	Per. Ident	Acc. Len	Accession
[x] gamma carbonic anhydrase family protein [Staphylococcus aureus]	Staphylococcus...	192	192	100%	1e-61	100.00%	97	NGG14433.1
[x] gamma carbonic anhydrase family protein [Salmonella enterica]	Salmonella ente...	192	192	100%	2e-61	98.97%	123	EGI1888674.1
[x] gamma carbonic anhydrase family protein [Salmonella enterica subsp. enterica serovar Typhimurium]	Salmonella ente...	192	192	100%	3e-61	98.97%	142	EDG6031664.1
[x] gamma carbonic anhydrase family protein [Salmonella enterica]	Salmonella ente...	193	193	100%	4e-61	98.97%	162	WP_162989120.1
[x] gamma carbonic anhydrase family protein [Salmonella enterica]	Salmonella ente...	193	193	100%	7e-61	98.97%	178	EFP6264598.1
[x] transferase [Salmonella sp. NCTC 11881]	Salmonella sp. ...	192	192	100%	7e-61	98.97%	156	VUC73367.1
[x] gamma carbonic anhydrase family protein [Salmonella enterica subsp. enterica serovar Sendai]	Salmonella ente...	193	193	100%	8e-61	98.97%	184	EBX8733770.1
[x] gamma carbonic anhydrase family protein [Salmonella enterica]	Salmonella ente...	193	193	100%	9e-61	98.97%	184	EDJ0681111.1
[x] gamma carbonic anhydrase family protein [Salmonella enterica]	Salmonella ente...	193	193	100%	9e-61	98.97%	184	EHD8855493.1
[x] gamma carbonic anhydrase family protein [Salmonella enterica]	Salmonella ente...	193	193	100%	9e-61	98.97%	184	WP_117335624.1
[x] gamma carbonic anhydrase family protein [Salmonella enterica subsp. enterica serovar Typhimurium]	Salmonella ente...	193	193	100%	9e-61	98.97%	184	TYN69919.1
[x] gamma carbonic anhydrase family protein [Salmonella enterica]	Salmonella ente...	193	193	100%	9e-61	98.97%	184	EBT7154459.1
[x] gamma carbonic anhydrase family protein [Salmonella enterica subsp. enterica serovar Typhimurium]	Salmonella ente...	193	193	100%	9e-61	98.97%	184	MBJ3343629.1
[x] gamma carbonic anhydrase family protein [Salmonella enterica subsp. enterica serovar Typhimurium]	Salmonella ente...	193	193	100%	1e-60	98.97%	184	ECS6664603.1
[x] Structure of the YrdA ferripyochelin binding protein from Salmonella enterica [Salmonella enterica subsp. en...	Salmonella ente...	193	193	100%	1e-60	98.97%	187	3R3R_A
[x] gamma carbonic anhydrase family protein [Salmonella enterica subsp. enterica serovar Typhimurium]	Salmonella ente...	193	193	100%	1e-60	98.97%	184	EBL5800546.1
[x] gamma carbonic anhydrase family protein [Salmonella enterica]	Salmonella ente...	193	193	100%	1e-60	98.97%	184	
[x] gamma carbonic anhydrase family protein [Salmonella enterica]	Salmonella ente...	193	193	100%	1e-60	98.97%	184	

3. >NGB42162.1 gamma carbonic anhydrase family protein [Staphylococcus aureus]

This protein sequence comes from an unpublished, direct submission to NCBI from *Staphylococcus aureus* strain UG302, unfinished genome with 167 contigs. **The submitter is the same as no.2.** According to PubMLST, the predicted taxa is 100% *S. aureus*. However, when we blasted this sequence in NCBI, we did not even get any hit against its own sequence or *S. aureus*. All of the 100 hits are from *Salmonella enterica*, with 99-100% identity.

Descriptions		Graphic Summary	Alignments	Taxonomy				
Sequences producing significant alignments								
Download New Select columns Show 100								
[ ] select all 0 sequences selected GenPept Graphics Distance tree of results Multiple alignment New MSA Viewer								
Description	Scientific Name	Max Score	Total Score	Query Cover	E value	Per. Ident	Acc. Len	Accession
[ ] Structure of the YrdA ferrityochelin binding protein from Salmonella enterica [Salmonella enterica subsp. e...]	Salmonella ente...	371	371	100%	1e-129	100.00%	187	3R3R_A
[ ] MULTISPECIES: gamma carbonic anhydrase family protein [Salmonella]	Salmonella	371	371	100%	2e-129	100.00%	184	WP_001285640.1
[ ] gamma carbonic anhydrase family protein [Salmonella enterica]	Salmonella ente...	370	370	100%	3e-129	99.46%	184	EBI7102533.1
[ ] TPA: gamma carbonic anhydrase family protein [Salmonella enterica]	Salmonella ente...	370	370	100%	3e-129	99.46%	184	HAD9512043.1
[ ] gamma carbonic anhydrase family protein [Salmonella enterica]	Salmonella ente...	370	370	100%	3e-129	99.46%	184	EEL2321507.1
[ ] TPA: gamma carbonic anhydrase family protein [Salmonella enterica]	Salmonella ente...	370	370	100%	3e-129	99.46%	184	HAK7698611.1
[ ] TPA: gamma carbonic anhydrase family protein [Salmonella enterica subsp. enterica serovar Typhimurium]	Salmonella ente...	370	370	100%	4e-129	99.46%	184	HAD0753013.1
[ ] MULTISPECIES: gamma carbonic anhydrase family protein [Salmonella]	Salmonella	370	370	100%	4e-129	99.46%	184	WP_000005248.1
[ ] gamma carbonic anhydrase family protein [Salmonella enterica]	Salmonella ente...	370	370	100%	4e-129	99.46%	184	WP_140770483.1
[ ] gamma carbonic anhydrase family protein [Salmonella enterica]	Salmonella ente...	370	370	100%	5e-129	99.46%	184	EGB2615634.1
[ ] gamma carbonic anhydrase family protein [Salmonella enterica subsp. enterica serovar Typhimurium]	Salmonella ente...	370	370	100%	5e-129	99.46%	184	ECS664603.1
[ ] gamma carbonic anhydrase family protein [Salmonella enterica subsp. enterica serovar Typhimurium]	Salmonella ente...	369	369	100%	5e-129	99.46%	184	EBX1672959.1
[ ] gamma carbonic anhydrase family protein [Salmonella enterica subsp. enterica serovar Typhimurium]	Salmonella ente...	369	369	100%	5e-129	99.46%	184	EHC3205005.1
[ ] gamma carbonic anhydrase family protein [Salmonella enterica]	Salmonella ente...	369	369	100%	5e-129	99.46%	184	EC07366945.1
[ ] gamma carbonic anhydrase family protein [Salmonella enterica subsp. enterica serovar Typhimurium]	Salmonella ente...	369	369	100%	5e-129	99.46%	184	EBL5800546.1
[ ] gamma carbonic anhydrase family protein [Salmonella enterica]	Salmonella ente...	369	369	100%	6e-129	99.46%	184	EBT7154459.1
[ ] gamma carbonic anhydrase family protein [Salmonella enterica]	Salmonella ente...	369	369	100%	6e-129	99.46%	184	

This protein sequence only appeared when the BLASTp is limited to *S. aureus*, which gave us exactly its own sequence and the 2 erroneous protein sequences listed here (NGB42162.1, NGG14433.1, WP_094666538.1).

Descriptions		Graphic Summary	Alignments	Taxonomy				
Sequences producing significant alignments								
Download New Select columns Show 100								
[x] select all 7 sequences selected GenPept Graphics Distance tree of results Multiple alignment New MSA Viewer								
Description	Scientific Name	Max Score	Total Score	Query Cover	E value	Per. Ident	Acc. Len	Accession
[x] gamma carbonic anhydrase family protein [Staphylococcus aureus]	Staphylococcus aureus	371	371	100%	6e-132	100.00%	184	NGB42162.1
[x] gamma carbonic anhydrase family protein [Staphylococcus aureus]	Staphylococcus aureus	193	193	52%	8e-63	98.97%	97	NGG14433.1
[x] gamma carbonic anhydrase family protein [Staphylococcus aureus]	Staphylococcus aureus	134	134	82%	5e-39	45.70%	149	WP_094666538.1
[x] MULTISPECIES: gamma carbonic anhydrase family protein [Staphylococcaceae]	Staphylococcaceae	107	107	92%	4e-28	39.18%	169	WP_037588601.1
[x] hypothetical protein [Staphylococcus aureus]	Staphylococcus aureus	94.4	94.4	54%	1e-23	45.10%	121	MBO8589741.1
[x] phenylacetic acid degradation protein PaaY [Staphylococcus aureus]	Staphylococcus aureus	57.8	57.8	48%	9e-10	33.33%	101	RZH80017.1
[x] acyltransferase [Staphylococcus aureus]	Staphylococcus aureus	38.1	38.1	61%	0.045	29.31%	172	WP_115294908.1

As with the earlier protein sequence which was submitted by the same group, **this protein sequence is not even from *S. aureus* but instead came from *Salmonella enterica*.**

4. WP_094666538.1 gamma carbonic anhydrase family protein, partial [*Staphylococcus aureus*].

This protein sequence comes from an unpublished, direct submission to NCBI from *Staphylococcus aureus* strain UV695, unfinished genome with 468 contigs. The

submitter is from another working group. According to PubMLST, the predicted taxa is 100% *S. aureus*. However, when we blasted this sequence in NCBI, we got almost all hits from *Enterococcus faecium* with 99-100% identity. This supposedly 'gamma CA from *S. aureus* is the only strain found among them' (highlighted in red box). Again, this clearly shows that **this sequence is not from *S. aureus* but from *Enterococcus faecium*.**

Descriptions		Graphic Summary	Alignments	Taxonomy				
Sequences producing significant alignments								
Download ▼ New Select columns ▼ Show 100 ▼ ?								
[x] select all 100 sequences selected GenPept Graphics Distance tree of results Multiple alignment New MSA Viewer								
Description	Scientific Name	Max Score	Total Score	Query Cover	E value	Per. Ident	Acc. Len	Accession
[x] TPA: gamma carbonic anhydrase family protein [Enterococcus faecium]	Enterococcus fa...	305	305	100%	3e-104	100.00%	159	HAQ8310572.1
[x] gamma carbonic anhydrase family protein [Enterococcus faecium]	Enterococcus fa...	304	304	100%	3e-104	100.00%	161	WP_125179327.1
[x] MULTISPECIES: gamma carbonic anhydrase family protein [Enterococcus]	Enterococcus	304	304	100%	3e-104	100.00%	161	WP_002289401.1
[x] TPA: gamma carbonic anhydrase family protein [Enterococcus faecium]	Enterococcus fa...	304	304	100%	3e-104	100.00%	156	HAQ8387424.1
[x] TPA: gamma carbonic anhydrase family protein [Enterococcus faecium]	Enterococcus fa...	304	304	100%	3e-104	100.00%	160	HAP9957657.1
[x] gamma carbonic anhydrase family protein [Enterococcus faecium]	Enterococcus fa...	304	304	100%	4e-104	100.00%	163	OSP60047.1
[x] gamma carbonic anhydrase family protein [Staphylococcus aureus]	Staphylococcus...	304	304	100%	4e-104	100.00%	149	WP_094666538.1
[x] TPA: gamma carbonic anhydrase family protein [Enterococcus faecium]	Enterococcus fa...	304	304	100%	4e-104	100.00%	155	HAQ8231580.1
[x] TPA: gamma carbonic anhydrase family protein [Enterococcus faecium]	Enterococcus fa...	303	303	100%	4e-104	100.00%	150	HAV0195165.1
[x] bacterial transferase hexapeptide repeat protein [Enterococcus faecium TX1330]	Enterococcus fa...	304	304	100%	7e-104	98.66%	180	EI61480.1
[x] gamma carbonic anhydrase family protein [Enterococcus faecium]	Enterococcus fa...	303	303	100%	7e-104	99.33%	161	EGP5337857.1
[x] MULTISPECIES: gamma carbonic anhydrase family protein [unclassified Enterococcus]	unclassified Ent...	303	303	100%	7e-104	99.33%	161	WP_008267279.1
[x] MULTISPECIES: gamma carbonic anhydrase family protein [Enterococcus]	Enterococcus	303	303	100%	7e-104	99.33%	161	WP_002319637.1
[x] gamma carbonic anhydrase family protein [Enterococcus faecium]	Enterococcus fa...	303	303	100%	7e-104	99.33%	161	WP_086324954.1
[x] MULTISPECIES: gamma carbonic anhydrase family protein [Enterococcus]	Enterococcus	303	303	100%	8e-104	98.66%	161	WP_002310223.1
[x] gamma carbonic anhydrase family protein [Enterococcus faecium]	Enterococcus fa...	303	303	100%	8e-104	98.66%	161	WP_123838076.1
[x] TPA: gamma carbonic anhydrase family protein [Enterococcus faecium]	Enterococcus fa...	303	303	100%	8e-104	99.33%	161	
[x] gamma carbonic anhydrase family protein [Enterococcus faecium]	Enterococcus fa...	303	303	100%	9e-104	98.66%	161	

In addition to the bioinformatic analysis, we also validated our claim with cloning experiments. If CA is present in *S. aureus*, the deletion of *mpsABC*, a bicarbonate transporter as the sole bicarbonate concentrating system in *S. aureus* would not result in such a severe growth phenotype. This phenotype is similar when CA is deleted from *S. carnosus*.

Based on our extensive analyses and experiments, we can confidently state that there is no CA present in the genome of *S. aureus* and thus there is no mistake in **line 134-135 and Table 1 & 3**. All the 4 other CAs sequences provided by reviewer #2 are all protein sequences that **belong to CAs from other bacteria and do not even exist in *S. aureus***. According to Aureowiki (AureoWiki-The repository of the Staphylococcus aureus research and annotation community by Fuchs et al., Int J Med Microbiol, 2018), there is no mention of carbonic anhydrase in its database. Aureowiki (<http://aureowiki.med.uni-greifswald.de>) is a **manually curated database** that provides detailed information on the genes and proteins of clinically and experimentally relevant *S. aureus* strains, currently covering NCTC 8325, COL, Newman, USA300_FPR3757, and N315.

Even, it was mentioned there is no CA in *S. aureus* in a "Nature Communication" paper that is not correct: Table 1 in "PMID: 31399577", which the first author and the corresponding author are similar to this manuscript. A big mistake was published in the "Nature Communication" in 2019 and another one has submitted to a journal from ASM now.

We did not deny that the publication PMID: 31399577 is not from our group. Our current manuscript clearly mentioned that this is a follow-up study of our previous work in PMID: 31399577 (line 137) and therefore the same authors appeared in the manuscript.

Our earlier paper PMID: 31399577 stated that the presence of CAs are inferred based on the occurrence of Pfam motif related to CA. In the present manuscript, we supplemented the presence of CA based on Pfam motif with similarity with CA from *S. carnosus*. As an example, we took the *Staphylococcus* strains listed in Table S1 and confirmed it with BLASTp of CA from *S. carnosus* (WP_015900702.1). As can be seen from the right column, the presence of the CAs based on Pfam motif correlate with the high percentage of identity with protein sequence of CA from *S. carnosus*.

Species	MpsAB	CA (based on Pfam)		Identity (%) with S. carnosus CA
		Pro	Euk	
Staphylococcus agnetis 908	-	+	-	73 (135/186)
Staphylococcus argenteus BN75	+	-	-	-
Staphylococcus aureus aureus MSHR1132	+	-	-	-
Staphylococcus aureus aureus USA300_FPR3757	+	-	-	-
Staphylococcus capitis AYP1020	+	-	-	-
Staphylococcus carnosus TM300	-	+	-	100 (192/192)
Staphylococcus cohnii SNUDS-2	+	-	-	-
Staphylococcus condimentii DSM 11674	-	+	-	97 (187/192)
Staphylococcus epidermidis RP62A	+	-	-	-
Staphylococcus equorum KS1039	+	-	-	-
Staphylococcus felis ATCC 49168	-	+	-	73 (135/186)
Staphylococcus haemolyticus JCSC1435	+	-	-	-
Staphylococcus hominis hominis K1	+	-	-	-
Staphylococcus hyicus ATCC 11249	-	+	-	72 (134/185)
Staphylococcus lugdunensis C_33	+	-	-	-
Staphylococcus lutrae ATCC 700373	-	+	-	70 (133/191)
Staphylococcus muscae NCTC 13833	-	+	-	70 (130/188)
Staphylococcus nepalensis JS1	+	-	-	-
Staphylococcus pasteurii SP1	+	-	-	-
Staphylococcus pettenkoferi FDAARGOS_288	-	+	-	75 (140/192)
Staphylococcus piscifermentans NCTC 13836	-	+	-	96 (185/192)

Staphylococcus pseudintermedius ED99	-	+	-	70 (131/188)
Staphylococcus saprophyticus 883	+	-	-	-
Staphylococcus schleiferi 1360-13	-	+	-	73 (132/185)
Staphylococcus sciuri SNUSD-18	(+)	+	-	67 (126/188)
Staphylococcus simiae NCTC 13838	+	-	-	-
Staphylococcus simulans FDAARGOS_124	-	+	-	87 (167/192)
Staphylococcus stepanovicii NCTC 13839	-	+	-	66 (125/190)
Staphylococcus succinus 14BME20	+	-	-	-
Staphylococcus warneri SG1	+	-	-	-
Staphylococcus xylosus SMQ121	+	-	-	-

We are aware of the following publications reporting the presence of CA in *S. aureus*.

1. Capasso, C. & Supuran, C. T. An overview of the alpha-, beta- and gamma-carbonic anhydrases from Bacteria: can bacterial carbonic anhydrases shed new light on evolution of bacteria? *J Enzyme Inhib Med Chem* 30, 325-332, doi:10.3109/14756366.2014.910202 (2015).

In this paper, it was mentioned that the genome of *S. aureus* encodes only for gamma CAs and this is also shown in Table 1 "for which the genome was cloned". However, no other information or citation given to support this statement.

2. Supuran, C. T. & Capasso, C. New light on bacterial carbonic anhydrases phylogeny based on the analysis of signal peptide sequences. *J Enzyme Inhib Med Chem* 31, 1254-1260, doi:10.1080/14756366.2016.1201479 (2016).

According to this paper, *S. aureus* was mentioned once in table 1 that listed it as having a gamma CA with protein EVX10196.1 and this sequence is used to build a phylogenetic tree. In the NCBI, EVX10196.1 is annotated as 2,3,4,5-tetrahydropyridine-2,6-dicarboxylate N-acetyltransferase from *S. aureus* M20916 which is an unfinished genome with 67 contigs. According to PubMLST, the predicted taxa is 100% *S. aureus*. This 239 amino acid protein is listed as non-essential by Aureowiki and is annotated as *dapD*, which is part of an operon consisting of 6 genes involved in the biosynthesis of lysine. Lysine is a particularly important amino acid in *S. aureus*, being required not only as a building block for proteins but also as a component of the cell wall peptidoglycan (Barbieri, J. T., et al., 2001. "Identification and Analysis of *Staphylococcus aureus* Components Expressed by a Model System of Growth in Serum." *Infection and Immunity* 69(8): 5198-5202). Therefore this protein EVX10196.1 is not a CA.

3. Urbanski, L. J. et al. Sulphonamide inhibition profile of *Staphylococcus aureus* beta-carbonic anhydrase. *J Enzyme Inhib Med Chem* 35, 1834-1839, doi:10.1080/14756366.2020.1826942 (2020).

This paper presents the production and kinetic and inhibitory characterization of beta CA from *S. aureus*. One of the co-author is the author of the above 2 papers which

earlier reported that *S. aureus* encodes **only gamma CA**. In this paper, the CA gene was obtained from UniProt, protein entry EZX15767 and was synthesized to produce a recombinant protein in *E. coli*. A search in UniProt revealed that this protein is encoded by a gene with locus tag V070_02709 from the *S. aureus* strain C0673. From our analysis above, **this strain C0673 is in fact *S. sciuri* and not *S. aureus*. Therefore, the CA activity described is from *S. sciuri*.**

4. Urbanski, L. J. et al. An anion and small molecule inhibition study of the β -carbonic anhydrase from *Staphylococcus aureus*, *J Enzyme Inhib Med Chem* 36:1, 1088-1092, DOI: 10.1080/14756366.2021.1931863 (2021).

This very recent paper by the same authors (no. 3) who followed up the previous study by reporting its inhibition profile with anions and other small molecules known to inhibit CAs. The same recombinant protein as no. 3 was described, which means that **they were still using the CA from C0673, which is actually *S. sciuri*.**

All these papers are published in the same journal (*J Enzyme Inhib Med Chem*). They are flawed and misleading and should be retracted.

8- Line 136: the CA from *Staphylococcus carnosus* belongs to what CA family that was mentioned in Figure 1 as well? Why? Please indicate in the manuscript.

We do not understand your comment here. "Line 135-137 merely state: On the other hand, a gene annotated as encoding a putative CA from *Staphylococcus carnosus* TM300 has yet to be investigated for its physiological function or enzymatic activity." We did, however, mention the CA family in **line 212**: "According to NCBI Conserved Domain database, the *S. carnosus* CA belongs to the β -class and D clade (cd03379)." Figure 1 is a multiple protein sequence alignment using the 2 strains which we used to in our experiment to generate CA mutants, *S. carnosus* and *S. pseudintermedius* along with CAs from *S. sciuri*, *Bacillus subtilis*, *Streptococcus pneumoniae* in which we showed the gene synteny in Supplementary Figure 1. We also included *E. coli* in the alignment because we used the *E. coli* CA mutant for our complementation experiments, meaning the CA from *E. coli* and *S. carnosus* is interchangeable.

In conclusion, among the almost 5000 *S. aureus* genomes that we searched for all the CA classes, there was in no case a convincing indication that there are such CAs present in *S. aureus* species. The statement in our "Nature Communication" paper is therefore correct. Furthermore, we have deleted *mpsAB* in 4 different well-characterized *S. aureus* strains. None of the *mpsAB* mutants could grow under ambient air, indicating that there is no functional CA. However, all publications in which a CA in *S. aureus* has been described and analyzed are dubious. We do not want to burden our current manuscript unnecessarily with the correction of these publications. We would address these flaws and mistakes in a short communication/letter to inform the scientific community about these inaccuracies in the journal and also the NCBI database.

June 30, 2021

Prof. Friedrich Götz
University of Tuebingen
Microbial Genetics, Interfaculty Institute of Microbiology and Infection Medicine (IMIT)
Auf der Morgenstelle 28
Tuebingen 72076
Germany

Re: Spectrum00305-21R1 (The MpsAB bicarbonate transporter is superior to carbonic anhydrase in biofilm-forming bacteria with limited CO₂ diffusion)

Dear Prof. Friedrich Götz:

After careful review of your responses the reviewers comments, your manuscript is now ready for acceptance. I have also consulted with members of our editorial board about your request to submit an accompanying short communication. We agree that there is merit in submitting a manuscript that corrects the literature regarding CA genes in *S. aureus*. However, we would consider this as a separate manuscript submission, which would be subject to peer review.

Your manuscript has been accepted, and I am forwarding it to the ASM Journals Department for publication. You will be notified when your proofs are ready to be viewed.

Sincerely,

Cezar Khursigara
Editor, Microbiology Spectrum
